# Bayesian Ensembling: Insights from Online Optimization and Empirical Bayes

**Daniel Waxman**                                                                         *dan@basis.ai*
*Stony Brook University**

**Fernando Llorente**                                                                  *fllorente@bnl.gov*
*Brookhaven National Laboratory*

**Petar M. Djurić**                                                      *petar.djuric@stonybrook.edu*
*Stony Brook University*

**Reviewed on OpenReview:** *https://openreview.net/forum?id=CCvVzmfBOn*

## Abstract

We revisit the classical problem of Bayesian ensembles and address the challenge of learning optimal combinations of Bayesian models in an online learning setting. To this end, we reinterpret existing approaches such as Bayesian model averaging (BMA) and Bayesian stacking through a novel empirical Bayes lens, shedding new light on the limitations and pathologies of BMA. Further motivated by insights from online optimization, we propose Online Bayesian Stacking (OBS), a method that optimizes the log-score over predictive distributions to adaptively combine Bayesian models. A key contribution of our work is establishing a novel connection between OBS and portfolio selection, bridging Bayesian ensemble learning with a rich, well-studied theoretical framework that offers efficient algorithms and extensive regret analysis. We further clarify the relationship between OBS and online BMA, showing that they optimize related but distinct cost functions. Through theoretical analysis and empirical evaluation, we identify scenarios where OBS outperforms online BMA and provide principled methods and guidance on when practitioners should prefer one approach over the other.

## 1 Introduction

Combining the opinions of multiple models is a pervasive problem in the statistical sciences, with many different names, approaches, and applications. In signal processing, for example, a commonly encountered problem is one of *sensor fusion*, where information reported from several different sensors must be combined to obtain the best possible estimate (Khaleghi et al., 2013). In econometrics, the problem is often known as *forecast combination*, spurred by the seminal work of Bates & Granger (1969). The Bayesian literature often refers to the problem as *opinion pooling*, whichreceived attention even in the early days of Bayesian statistics (Stone, 1961; DeGroot, 1974; Genest & Zidek, 1986). While many of the above approaches originated in the combination of point estimates, the combination of probability distributions has received significant attention since (Koliander et al., 2022).

In machine learning, this issue is commonly known as *ensembling*, with much recent interest due to the development of diverse models, architectures, and training modalities (Dietterich, 2000). Many ensembling approaches exist, hinging on different assumptions and asymptotic guarantees. We will restrict our attention to Bayesian machine learning, where the goal is to combine the estimates of $K$ different *probabilistic* models $\mathcal{M}_1, \ldots, \mathcal{M}_K$. The classical approach in this setting, known as *Bayesian model averaging (BMA)*, is to weigh the estimates of each model according to their marginal likelihood (Hoeting et al., 1999). When the data were

---

*Work done at Stony Brook University. Current address: Basis Research Institute `<dan@basis.ai>`.

generated by one of the models $\mathcal{M}_1, \ldots, \mathcal{M}_K$, this is the "correct" way to combine models from a Bayesian perspective and is optimal, in the sense of choosing the correct model with probability 1 in the limit of infinite data (Yao et al., 2018).

More recently, Yao et al. (2018) critically examined BMA in a more typical setting where the data are not generated by any of the candidate models. They instead proposed *Bayesian stacking*, which optimizes a log-score criterion; asymptotically, Bayesian stacking corresponds to choosing the optimal *convex combination* of models, juxtaposed with the optimal singular model selected by BMA. Similar discussions and proposals have previously appeared in the econometrics community for forecasting models (Hall & Mitchell, 2007; Geweke & Amisano, 2012).

In this paper, we further study the online stacking problem, introducing a novel analysis via empirical Bayes before moving to a new analysis of Bayesian stacking in the online setting. We summarize our contributions as follows:

1. We show that *Online Bayesian Stacking (OBS)* corresponds exactly to the well-studied problem of (universal) online portfolio selection (OPS). This connection allows us to reinterpret Bayesian ensemble learning through the lens of online convex optimization and to leverage efficient, theoretically grounded algorithms (e.g., Exponentiated Gradient and the Online Newton Step).

2. We show theoretically how one can choose between OBS and online BMA with constant regret. Building on this connection, we discuss how regret bounds from the OPS literature can be applied to OBS.

3. We introduce a simple yet compelling argument via empirical Bayes to explain why BMA collapses and how Bayesian stacking aids in avoiding this common problem.

4. We perform an extensive empirical analysis of OBS using ensembles of state-of-the-art models, including Gaussian processes, variational Bayesian neural networks, and stochastic volatility forecasting models estimated with sequential Monte Carlo. We show that, in all cases tested, OBS significantly outperforms online BMA (O-BMA) and dynamic model averaging (DMA), with additional computational cost that is often negligible in comparison to online training/prediction of ML models. We also show that OBS can be particularly beneficial in non-stationary environments, often outperforming any set of fixed weights.

The rest of this article is structured as follows: in Section 2, we review the problem of Bayesian ensembling, discussing the BMA and stacking approaches. In Section 3, we show how the online variant of Bayesian stacking corresponds to the well-known (universal, online) portfolio selection problem from online convex optimization, and discuss corresponding insights that show OBS's relation to O-BMA. In Section 4, we exploit this connection to derive a novel comparison of BMA and Bayesian stacking from an empirical Bayes perspective. In Section 5, we carry out a number of realistic experiments on synthesized and real data before finally concluding in Section 6.

## 2    Bayesian Ensembles: A Review of Existing Methods

In this section, we review the ideas of BMA and Bayesian stacking Yao et al. (2018) and discuss some related work.

### 2.1    Bayesian Model Averaging

Bayesian ensembling methods combine a set of Bayesian models for predictive inference. We will focus on methods that create linear mixtures of posterior predictive distributions: given models $\mathcal{M}_1, \ldots, \mathcal{M}_K$ each mapping $\mathbf{x} \in \mathcal{X} \subseteq \mathbb{R}^d$ to a probability (density) over $y \in \mathcal{Y} \subseteq \mathbb{R}^r$, trained on a dataset $\mathcal{D}$, the task of Bayesian ensembling is thus to find a weight vector $\mathbf{w}$ in the $(K-1)$-dimensional simplex $\mathbb{S}^K$, which induces a posterior predictive distribution

$$p^{\text{ens}}(y \mid \mathbf{x}, \mathcal{D}) := \sum_k w_k p_k(y \mid \mathbf{x}, \mathcal{D}), \tag{1}$$

where $p_k(y \mid \mathbf{x}, \mathcal{D})$ is the posterior predictive distribution of model $\mathcal{M}_k$ on the dataset $\mathcal{D}$.

The most common method for combining Bayesian models is BMA (Hoeting et al., 1999), which forms the weight for a linear mixture of predictive distributions by simulating a posterior probability for model $\mathcal{M}_k$ given data $\mathcal{D}$ for weighting,

$$w_k^{\text{BMA}} \coloneqq \Pr(\mathcal{M}_k \mid \mathcal{D}) = \frac{p_k(\mathcal{D}) \Pr(\mathcal{M}_k)}{\sum_k p_k(\mathcal{D}) \Pr(\mathcal{M}_k)}, \tag{2}$$

where $p_k(\mathcal{D}) \coloneqq p(\mathcal{D}|\mathcal{M}_k)$ is the evidence of $\mathcal{M}_k$, and $\Pr(\mathcal{M}_k)$ is the prior probability of $\mathcal{M}_k$.

The appeal of BMA to a Bayesian is immediate: if $\Pr(\mathcal{M})$ has support over the data-generating distributions, then the resulting mixture Equation (1) is a straightforward application of Bayes' rule and is thus optimal in an information-theoretic sense (Zellner, 1988).

Much of our work focuses on the *online* setting, where BMA has several additional advantages. In this setting, we first obtain a new data point $\mathbf{x}_{t+1}$, with which we must make a prediction using the available data $\mathcal{D}_t$. After making a prediction, the value $y_{t+1}$ is revealed and the dataset $\mathcal{D}_{t+1} \coloneqq \mathcal{D} \cup \{\mathbf{x}_{t+1}, y_{t+1}\}$ is updated. In this case, we can compute $w_k^{\text{BMA}}$ recursively using posterior predictive distributions, which allows us to apply *exact* BMA at time $t$ using the currently-available information: denoting the weights at time $t$ as $w_{t,k}$, with $w_{0,k} = \Pr(\mathcal{M}_k)$, (2) becomes

$$w_{t+1,k}^{\text{BMA}} = \frac{w_{t,k} p_k(y_{t+1} \mid \mathbf{x}_{t+1}, \mathcal{D}_t)}{\sum_k w_{t,k} p_k(y_{t+1} \mid \mathbf{x}_{t+1}, \mathcal{D}_t)}. \tag{3}$$

Two main problems arise when using BMA: first, estimation via Equation (2) or Equation (3) requires access to predictive distributions, either through the marginal likelihood (i.e., the prior predictive) or the posterior predictive. This is often a surmountable problem, as marginal likelihoods are usually available in conjugate models, and BMA with approximated marginal likelihoods has also performed well empirically (Gómez-Rubio et al., 2020).

Perhaps more importantly, BMA is only optimal in terms of predictive error in the so-called "M-closed" setting, where the data were generated by one of the models $\mathcal{M}_1, \ldots, \mathcal{M}_K$ (Yao et al., 2018; Minka, 2000). Indeed, in the limit of infinite data, BMA weights concentrate on the single model that most closely resembles the data-generating process. We provide a novel empirical Bayes-based analysis of this fact in Section 4. This may result in arbitrarily poor posterior predictive accuracy of an ensemble using $\mathbf{w}^{\text{BMA}}$ relative to some other set of weights $\mathbf{w}^*$. Additionally, O-BMA can numerically collapse to the "wrong" model, never recovering due to numerical underflow (Waxman & Djurić, 2024).

## 2.2 Bayesian Stacking

Bayesian stacking (Yao et al., 2018) (similarly explored by (Clyde & Iversen, 2013; Le & Clarke, 2017)) presents an alternative way to derive a weight vector $\mathbf{w}$. In particular, Bayesian stacking finds the optimal weight vector $\mathbf{w}^*$ in the $(K-1)$-dimensional simplex $\mathbb{S}^K$ by maximizing some score $\mathcal{S}(\mathbf{w}, \mathcal{D}')$ over a separate dataset $\mathcal{D}'$:

$$\mathbf{w}^* \coloneqq \underset{\mathbf{w} \in \mathbb{S}^K}{\arg\max} \, \mathcal{S}(\mathbf{w}, \mathcal{D}'). \tag{4}$$

Particular attention is given to the log-score; for the predictive dataset (i.e., a holdout or validation set) $\mathcal{D}' = \{(\mathbf{x}_n, y_n)\}_{n=1}^N$[1], the corresponding optimization problem is

$$\mathbf{w}^* \coloneqq \underset{\mathbf{w} \in \mathbb{S}^K}{\arg\max} \sum_{n=1}^N \log \sum_{k=1}^K w_k p_k(y_n \mid \mathbf{x}_n, \mathcal{D}), \tag{5}$$

where $p_k(y_n \mid \mathbf{x}_n, \mathcal{D})$ is the predictive distribution of model $\mathcal{M}_k$. This can be recognized as finding the mixture of estimators with the highest predictive likelihood over $\mathcal{D}'$. This, in turn, minimizes the KL divergence

---

[1]We will typically use $n = 1, \ldots, N$ to denote datasets that are not processed sequentially/online, and $t = 1, \ldots, T$ for their sequential/online counterparts.

between the mixture $q(y \,|\, \mathbf{x}) = \sum_k w_k p_k(y \,|\, \mathbf{x}, \mathcal{D})$ and the predictive distribution, $p(y \,|\, \mathbf{x})$, which represents the (unknown) true generating mechanism of $y$ (Yao et al., 2018).

The optimization problem (5) is convex and provides nice asymptotic guarantees, but "wastes" data by requiring two separate datasets $\mathcal{D}$ and $\mathcal{D}'$. Yao et al. (2018) address this by showing the score in (5) is well-approximated by the leave-one-out (LOO) predictive density $p(y_n \,|\, \mathbf{x}_n, \mathcal{D}_{-n})$, where $\mathcal{D}_{-n} = \mathcal{D} \setminus \{(\mathbf{x}_n, y_n)\}$. For many models, the LOO predictive density can be efficiently estimated with Pareto-smoothed importance sampling (Vehtari et al., 2024), resulting in an efficient optimization problem with a single dataset $\mathcal{D}$. The resulting mixture of estimators has empirically shown performance superior to that of other combination methods, leading to several recent applications.

### 2.3  Bayesian Stacking for Time Series & Online Bayesian Stacking

In the published discussion to Yao et al. (2018), Ferreira (2018) discusses Bayesian stacking for time series, where forecasting densities naturally serve as predictive distributions, i.e., (5) becomes

$$\mathbf{w}^* \coloneqq \underset{\mathbf{w} \in \mathbb{S}^K}{\arg\max} \sum_{t=t^*+1}^{T} \log \sum_{k=1}^{K} w_k p_k(y_t \,|\, \mathbf{x}_t, \mathcal{D}_{1:t-1}), \tag{6}$$

where the first $t^*$ data are devoted to (pre-)training each model, and $\mathcal{D}_{1:t-1}$ denotes the first $t-1$ data points. This has notable similarities to pooling methods described by Hall & Mitchell (2007); Geweke & Amisano (2011), with the exception of the summation starting with index $t = 1$, reminiscent of the LOO approach. This method was applied by Geweke & Amisano (2012), where optimal weights were computed quarterly. "Windowed" approaches for autocorrelated time series data, where only the last $T_p$ points are considered, have also been deployed (Jore et al., 2010; Aastveit et al., 2014), but these are suboptimal in the more general case where data may be exchangeable conditioned on $\mathbf{x}_t$. Related to windowed approaches is dynamic model averaging, which is O-BMA with forgetting factors (Raftery et al., 2010). To the best of our knowledge, Bayesian stacking and its time series variants have not been applied in the *online* setting, where weights are estimated as new data become available. We will refer to approaches to the online problem as *online Bayesian stacking (OBS)*.

As we will see, OBS is a special case of online convex optimization (OCO) (Hazan, 2022), where "learning from experts" is well-studied. We show that the popular Hedge algorithm (Freund & Schapire, 1997) generalizes BMA with a learning rate, but optimizes a different loss function from OBS. Our proposed OBS is differentiated from existing approaches in several ways. First, it directly emulates Bayesian stacking, of recent interest to the Bayesian community. Second, our connection to OCO yields an extremely efficient implementation, unlike methods based on data-driven portfolio selection (e.g., Baştürk et al. (2019)) that rely on particle filtering. Finally, our approach is more general for online learning, whereas data-driven strategies rely on the autocorrelation of time series data.

## 3  Insights from Online Convex Optimization

As Bayesian stacking is natively framed as an optimization problem, it is natural to study OBS from the optimization perspective as well. It turns out that our studies here are fruitful: by interpreting posterior predictive values as asset prices, OBS corresponds exactly to a classical problem of portfolio selection. An OCO perspective additionally rediscovers a development of Vovk (2001), which shows that BMA is exactly the Hedge algorithm with a specific choice of learning rate. We additionally discuss a hybrid approach meant to detect whether the model class is M-open or M-closed (or, more precisely, if OBS mixtures will outperform expert selection).

### 3.1  The Portfolio Selection Problem

We will first review the basic ideas of online portfolio selection (OPS), but our discussion will necessarily be brief; the interested reader is referred to the recent survey of Li & Hoi (2014), or the excellent lecture notes of Hazan (2022); Orabona (2019).

**Problem Statement** Consider an investment manager overseeing $K$ stocks, aiming to maximize total future wealth. They create the *best constantly rebalanced portfolio (BCRP)*, i.e., at each time step they re-allocate current wealth such that $w_k$ is in stock $k$ (classic portfolio selection assumes no transaction fees). Assets are allocated based on *price-relative vectors* $\mathbf{r}_t$, where $r_{t,k}$ is stock $k$'s relative value increase from day $t - 1$ to $t$. The goal is to maximize the (multiplicative) wealth at time $T$, or more conveniently, the (additive) log-wealth:

$$\text{wealth}_T := \prod_{t=1}^{T} \mathbf{w}^\top \mathbf{r}_t \iff \log \text{wealth}_T := \sum_{t=1}^{T} \log(\mathbf{w}^\top \mathbf{r}_t). \tag{7}$$

In the *online* version of the portfolio selection problem, we seek to find $\mathbf{w}_t$ that minimizes the regret with respect to the optimal weights $\mathbf{w}^*$,

$$R_T := \sum_{t=1}^{T} \log(\mathbf{w}^{*\top} \mathbf{r}_t) - \sum_{t=1}^{T} \log(\mathbf{w}_t^\top \mathbf{r}_t), \tag{8}$$

where the weights $\mathbf{w}_t$ are determined after observing $\mathbf{r}_{t-1}$.

Although many approaches explicitly model returns as stochastic processes, others allow $r_{t,k}$ to be chosen arbitrarily, and indeed by an adversary. Algorithms that achieve sublinear regret with respect to any non-negative sequence of returns $r_{t,k}$ are known as *universal* (Orabona, 2019, p. 144). We draw an analogy between the price-relative vectors $\mathbf{r}_t$ and the predictive likelihoods $p(y_t \,|\, \mathbf{x}_t, \mathcal{D}_{t-1})$, which make the adversarial case of particular interest to us.

**Algorithms** The portfolio selection problem has an attractive structure from the optimization perspective: the log wealth is a concave function defined over the simplex, a convex set. OPS therefore falls under the (online) convex optimization (OCO) umbrella (Hazan, 2022; Orabona, 2019). To keep with standard terminology, we will introduce each method as minimizing the convex loss function given by $-\log \text{wealth}_T$, which is equivalent to maximizing the concave function $\log \text{wealth}_T$.

The simplest OCO algorithm is online (sub)gradient descent (OGD), a straightforward extension of the classical gradient descent algorithm to a sequence of losses $\ell_1, \ldots, \ell_T$, each a function of some generic quantity $\boldsymbol{\theta}$ belonging to a convex set $\mathcal{K}$. In our case, $\boldsymbol{\theta}_t$ represents weights $\mathbf{w}_t$, $\ell_t$ is the negative log-return, $-\log(\mathbf{w}_t^\top \mathbf{r}_t)$, and $\mathcal{K}$ is the simplex $\mathbb{S}^K$. At each time instant, $\boldsymbol{\theta}_t$ is updated in the direction of the gradient $\nabla \ell_t$ and then projected back onto the convex set $\mathcal{K}$ (e.g., (Hazan, 2022, Algorithm 2.3)).

In many problems, it turns out to be beneficial to consider regularizers that account for the geometry of $\mathcal{K}$ explicitly, resulting in different update steps. This is an online form of the classical mirror descent (Nemirovsky et al., 1983). When $\mathcal{K}$ is the simplex $\mathbb{S}^K$ (as in OPS), one popular choice is entropic regularization (e.g., (Hazan, 2022, Section 5.4.2)). The resulting algorithm is known as *Exponentiated Gradients (EG)* (Helmbold et al., 1998), which uses the update step

$$\mathbf{w}_{t+1} = \frac{\mathbf{w}_t \odot \exp(-\eta \nabla_\mathbf{w} \ell_t)}{\|\mathbf{w}_t \odot \exp(-\eta \nabla_\mathbf{w} \ell_t)\|_1}, \tag{9}$$

where $\odot$ is the elementwise product, and $\eta$ is a learning rate. For the cost function $\ell_t(\mathbf{w}) = -\log \mathbf{w}_t^\top \mathbf{r}_t$, we have $\nabla_\mathbf{w} \ell_t = -\mathbf{r}_t / \mathbf{w}_t^\top \mathbf{r}_t$, and therefore,

$$\mathbf{w}_{t+1} = \frac{\mathbf{w}_t \odot \exp(\eta \mathbf{r}_t / \mathbf{w}_t^\top \mathbf{r}_t)}{\|\mathbf{w}_t \odot \exp(\eta \mathbf{r}_t / \mathbf{w}_t^\top \mathbf{r}_t)\|_1}. \tag{10}$$

The EG algorithm is conceptually simple, with relatively good regret bounds whenever the gradient is bounded. However, more modern algorithms, such as the *online Newton step (ONS)* (Hazan et al., 2007), can provide tighter bounds on the regret.

Of note for applications in non-stationary environments are OCO algorithms designed for such environments. In our experiments, for example, we will utilize *discounted ONS (D-ONS)* (Yuan & Lamperski, 2020), an ONS variant that includes a forgetting factor over the second-order information.

---

**Algorithm 1** Online Ensemble Update: **OBS with EG** or **OBS with Soft-Bayes** or **O-BMA**.

---

1: **Input:** Data stream $\{(\mathbf{x}_t, y_t)\}_{t=1}^T$, models $\{M_k\}_{k=1}^K$, initial weights $\mathbf{w}_0 \in \Delta^K$, learning rate $\eta$.
2: **for** $t = 1$ **to** $T$ **do**
    *// Prediction Step*
3:    Receive $\mathbf{x}_t$ and compute each model's predictive density $p_k(y \mid \mathbf{x}_t, \mathcal{D}_{t-1})$ for $k = 1, \ldots, K$
4:    Form ensemble prediction via Equation (1)
5:    Output prediction and receive true label $y_t$.
    *// Update Step*
6:    Update model weights using **OBS w/ EG** (10) or **OBS w/ Soft-Bayes** (11) or **O-BMA** (3)
7:    Update dataset: $\mathcal{D}_t \leftarrow \mathcal{D}_{t-1} \cup \{(\mathbf{x}_t, y_t)\}$.
8: **end for**

---

**Market Variability and Soft-Bayes** In order to achieve optimal regret bounds with EG or ONS, we must assume that the ratio $\alpha$ of the minimum return to maximum return at any time $t$ — called the *market variability parameter* — is bounded. In this case, $\ell_t$ is 1-exp-concave with bounded gradients, meaning that $\exp(-\ell_t) = \mathbf{w}_t^\top \mathbf{r}_t$ is a concave function (Hazan, 2022). So long as this assumption holds, the ONS algorithm provides tighter regret bounds than EG.

Moving beyond the assumption of a bounded $\alpha$ is the more recent Soft-Bayes (Orseau et al., 2017). Soft-Bayes proposes weight updates using a learning rate $\eta \in (0, 1)$ as

$$\mathbf{w}_{t+1} = \mathbf{w}_t \odot \left( 1 - \eta + \eta \frac{\mathbf{r}_t}{\mathbf{w}_t^\top \mathbf{r}_t} \right). \tag{11}$$

Orseau et al. (2017) provide an interpretation of Soft-Bayes in terms of "slowing down" O-BMA, and the resulting algorithm provides state-of-the-art regret bounds without the assumption of bounded gradients. We remark that while Soft-Bayes is in some aspects similar to our work, their focus is on developing algorithms and theoretical bounds for log-loss mixtures of experts. In the current work, we focus on statistical insights, connecting this online log-loss problem to the recently popular Bayesian stacking, and showing empirically that OBS is viable for modern Bayesian machine learning.

## 3.2 Online Bayesian Stacking is Portfolio Selection

Our core insight in this section is that the utility of (6) becomes (7) when the market gain $r_{t,k}$ is defined by the predictive density $p_k(y_t \mid \mathbf{x}_t, \mathcal{D}_{t-1})$. Indeed, our only requirement in a universal portfolio algorithm is that $r_{t,k}$ is nonzero, which is a rather mild and near-universally satisfied assumption for the predictive distribution of a regression model. Furthermore, using a constant rebalanced portfolio in the regret (8) is appropriate, as a constant rebalanced portfolio corresponds to the constant weights used in (offline) Bayesian stacking.

A point of nuance in applying OPS algorithms is the existence of a market variability parameter $\alpha$, requiring pre-determined maximum and minimum predictive densities. The maximum is clear for proper Bayesian models, and a minimum may be assumed (e.g., via compact data spaces or bounded model error). However, $\alpha$ may still be very small, which can be problematic for EG/ONS regret analysis. Bounding regret using subgradient norms can produce tighter guarantees, since non-adversarial predictions from failing models (i.e., $w_{t,k} \approx 0$) are unlikely to suddenly dominate. If such outliers are a concern, Soft-Bayes also provides $\alpha$-independent regret bounds.

We provide pseudocode for OBS and O-BMA in Algorithm 1. The pseudocode underscores the similarity between OBS and O-BMA, replacing a single update equation. Other algorithms for OPS may be used by changing Line 6.

## 3.3 Online Bayesian Model Averaging is the Hedge Algorithm with Learning Rate 1

Now that we have established an equivalence between OBS and the portfolio selection problem, one may wonder if a similar connection holds for O-BMA. Indeed, it is classically known that O-BMA updates are a

specific choice of the *Hedge algorithm* (Freund & Schapire, 1997), which performs expert selection. While this connection is established (Vovk, 2001), we rederive it as a useful narrative tool to motivate O-BMA as a model selection algorithm, which we show clearly using Proposition 3.1.

Close inspection reveals that $w_{t+1,k}^{\mathrm{BMA}}$ is precisely recovered by EG when the learning rate is $\eta = 1$ and the loss is

$$\ell_t(\mathbf{w}_t) = -\sum_k w_{t,k} \log p_k(y_t \mid \mathbf{x}_t, \mathcal{D}_{t-1}), \tag{12}$$

whose gradient is $\nabla_{w_{t,k}} \ell_t(\mathbf{w_t}) = -\log p_k(y_t|\mathbf{x}_t, \mathcal{D}_{t-1})$. This suggests that O-BMA may act similarly to an OCO algorithm that minimizes regret with respect to a different loss function,

$$\mathcal{L} = -\sum_{t=1}^{T} \sum_{k=1}^{K} w_k \log p_k(y_t \mid \mathbf{x}_t, \mathcal{D}_{t-1}). \tag{13}$$

This reveals a key insight: OBS aims for the best post-hoc expert *mixture*, while O-BMA targets the best *single* expert. The corresponding bound achieves constant regret; in fact, in the typical proof of this regret upper bound, we prove an equality regarding the regret. The following result is therefore known, but is presented in a somewhat unorthodox way to allow us to show *lower* bounds on the regret.

**Proposition 3.1.** *Let the regret of the BMA mixture with respect to the best individual model be defined as* $R_T = \sum_t \log p_{k^*}(y_t \mid \mathbf{x}_t, \mathcal{D}_{t-1}) - \sum_t \log \left( \sum_k w_{t,k} p_k(y_t \mid \mathbf{x}_t, \mathcal{D}_{t-1}) \right)$, *where* $k_*$ *is the model with the largest marginal likelihood. Then* $R_T$ *is related to an evidence gain in* $M_{k^*}$,

$$R_T = \log \mathrm{Pr}(\mathcal{M}_{k^*} \mid \mathcal{D}_T) / \mathrm{Pr}(\mathcal{M}_{k^*}). \tag{14}$$

Thus, if $\mathrm{Pr}(\mathcal{M}_{k^*} \mid \mathcal{D})$ is bounded below, so is the regret. In particular, Proposition 3.1 applied the following theorem directly, which asserts O-BMA acts as an "optimizer" to Equation (13).

**Corollary 3.2.** *Let the regret of the BMA mixture with respect to the best individual model be defined as in Proposition 3.1. If the posterior probability of the optimal model* $\mathcal{M}_{k^*}$ *exceeds its prior probability, i.e.,* $\mathrm{Pr}(\mathcal{M}_{k^*} \mid \mathcal{D}) \geq \mathrm{Pr}(\mathcal{M}_{k^*})$, *then* $R_T$ *is bounded both above and below,*

$$0 \leq R_T \leq -\log \mathrm{Pr}(\mathcal{M}_{k^*}). \tag{15}$$

Thus, under typical scenarios (e.g., uniform priors on $\mathcal{M}_k$), Equation (15) becomes fairly tight, and the solutions become strong online optimizers of Equation (15).

### 3.4 Regret Analysis

Connecting OBS with OPS makes regret bounds from the OCO literature available. Different choices of algorithms and assumptions on the predictive densities $\mathbf{p}_t$ affect the obtainable regret analysis (c.f. Van Erven et al. (2020, Table 1)); however, without additional assumptions on $\mathbf{p}_t$, the most efficient algorithms typically obtain regret of order $\mathcal{O}(\sqrt{T})$ (such as Soft-Bayes (Orseau et al., 2017)). If a bound on the norm of the gradients may be assumed, ONS can provide regret on the order $\mathcal{O}(\log T)$ (with runtime scaling with $K^2$). Variants of simple OCO algorithms may even provide regret with respect to time-varying optimal parameters, such as the D-ONS algorithm (Yuan & Lamperski, 2020).

We further discuss regret bounds in Appendix A. In Appendix A.3, we provide an example that shows that, with mild additional assumptions, we can even recover regret bounds from O-BMA applications, albeit with potentially worse constants - this is despite the fact that O-BMA cannot provide comparable bounds when regret is measured against the mixture loss.

### 3.5 A Hybrid Approach

In the case where the M-closed scenario is plausible, but this fact is unknown, OBS still achieves vanishing regret with respect to the singular best expert: this follows because the problem of determining an optimal

mixture of densities is harder than the corresponding expert selection problem, which is included as a special case.

Nevertheless, as determined in Proposition 3.1, the corresponding regret guarantees for O-BMA in the M-closed setting are extremely strong, achieving constant regret as an upper bound. When a model extremely close to the data generating process is available, OBS is thus potentially suboptimal.

One potential approach to ameliorate this is to consider a hybrid method, where O-BMA and OBS mixtures are maintained and further averaged via a secondary layer of O-BMA. That is, we consider two sets of weights, $\mathbf{w}_t^{\text{O-BMA}}$ and $\mathbf{w}_t^{\text{OBS}}$, along with a secondary set of weights $\mathbf{v}_t \in \mathbb{S}^2$. The corresponding predictive distribution is thus

$$p^{\text{hybrid}}(y_t \,|\, \mathbf{x}_t, \mathcal{D}_{1:t-1}) = v_1 \underbrace{\left( \sum_k w_{t,k}^{\text{O-BMA}} p_k(y_t \,|\, \mathbf{x}_t, \mathcal{D}_{t-1}) \right)}_{p^{\text{O-BMA}}(y_t \,|\, \mathbf{x}_t, \mathcal{D}_{t-1})} + v_2 \underbrace{\left( \sum_k w_{t,k}^{\text{OBS}} p_k(y_t \,|\, \mathbf{x}_t, \mathcal{D}_{t-1}) \right)}_{p^{\text{OBS}}(y_t \,|\, \mathbf{x}_t, \mathcal{D}_{t-1})}. \tag{16}$$

The predictive performance of this mixture achieves constant regret with respect to the best model, once again, while maintaining the expressiveness of the corresponding OBS solution. In particular, we may obtain the following bound:

**Theorem 3.3.** *Let the regret of the hybrid mixture in Equation* (16) *for the best individual model be defined as in Proposition 3.1, i.e.,* $R_T = \sum_t \log p_{k^*}(y_t \,|\, \mathbf{x}_t, \mathcal{D}_{t-1}) - \sum_t \log p^{hybrid}(y_t \,|\, \mathbf{x}_t, \mathcal{D}_{t-1})$, *where* $k^*$ *is the model with the largest marginal likelihood. Further assume uniform prior weights over* $\mathbf{v}$ *and* $\mathbf{w}^{O\text{-}BMA}$*. Then* $R_T$ *may be bounded as*

$$R_T \leq \log K + \log 2. \tag{17}$$

*Proof.* From the "outermost" perspective of ensembling (i.e., with $\mathbf{v}$), we have by Proposition 3.1 a regret bound with respect to any "outer" expert (i.e., the O-BMA or OBS mixtures):

$$\sum_t \log p^{\text{O-BMA}}(y_t \,|\, \mathbf{x}_t, \mathcal{D}_{t-1}) - \sum_t \log p^{\text{hybrid}}(y_t \,|\, \mathbf{x}_t, \mathcal{D}_{t-1}) \leq \log 2. \tag{18}$$

Further, from Proposition 3.1, O-BMA achieves a bounded regret

$$\sum_t \log p_{k^*}(y_t \,|\, \mathbf{x}_t, \mathcal{D}_{t-1}) - \sum_t \log p^{\text{O-BMA}}(y_t \,|\, \mathbf{x}_t, \mathcal{D}_{t-1}) \leq \log K. \tag{19}$$

Combining Equation (18) and Equation (19) completes the theorem. □

We note by a symmetric argument that if the OBS mixture has a clearly stronger predictive likelihood, the "outer" level of O-BMA will select the OBS mixture as the best expert, similarly incurring a simple additive regret factor.

## 4 An Empirical Bayes Perspective

An empirical Bayes (EB) perspective (Robbins, 1964) elucidates BMA and Bayesian stacking properties, offering direct arguments for their limitations. Recall that in EB, hyperparameters $\psi$ are found by maximizing the marginal likelihood, rather than being marginalized out. We show that O-BMA and OBS have subtly distinct EB justifications. While we focus notationally on the online setting, similar results hold for batch processing.

Let $p(y_1|\psi)$ be the prior predictive density of $y_1$ (model parameters are integrated out), and $p(y_i|\mathbf{y}_{1:i-1}, \psi)$ for $i = 2, \ldots, t$ are posterior predictive densities. Then the *prequential principle* (Dawid, 1984) studies a model through a predictive decomposition, choosing $\psi$ via $p(\mathbf{y}_{1:t}|\psi) = \prod_\tau p(y_\tau|\mathbf{y}_{1:\tau-1}, \psi)$. Further selecting $\psi$ through the optimization problem

$$\psi_\star = \arg\max_\psi \sum_{i=1}^t \log p(y_i|\mathbf{y}_{1:i-1}, \psi) \tag{20}$$

is termed *empirical Bayes* or type-II maximum likelihood estimate.

### 4.1 BMA is Empirical Bayes over an Indicator Variable

It is well-known in the literature that BMA collapses to the model with the highest marginal likelihood when the amount of data increases (Yao et al., 2018), but explanations of this fact are not often presented. Using our insights that O-BMA is the Hedge algorithm, i.e., that it minimizes regret with the loss function (13), this fact is straightforward to see: $w_k$ should be 1 for the model with the highest marginal likelihood and 0 for the others. In this sense, we may interpret BMA as performing empirical Bayes over an indicator variable $\mathbf{z}$ in the corresponding mixture distribution. To see this, we note that (13) can be written as

$$\mathcal{L} = \sum_{k=1}^{K} w_k \log \prod_{t=1}^{T} p_k(y_t \mid \mathbf{x}_t, \mathcal{D}_{t-1}) = \sum_{k=1}^{K} w_k \log p_k(\mathcal{D}_T), \tag{21}$$

and that $\sum_{k=1}^{K} w_k \log p_k(\mathcal{D}_T) \leq \max_k \log p_k(\mathcal{D}_T)$. Hence, the optimal weight vector corresponds to $w_k = 0$ for all $k$ except for the model $k^* = \arg\max_k p_k(\mathcal{D}_T)$ with the highest marginal likelihood, for which $w_{k^*} = 1$. Note also that by Jensen's inequality, (21) is a lower bound to $\log(\sum_k w_k p_k(\mathcal{D}_T))$. If we set $w_k = \Pr(\mathcal{M}_k)$, this is the log of the evidence of the BMA model, so we can interpret BMA as performing empirical Bayes on the discrete prior probabilities of the models.

### 4.2 OBS is Empirical Bayes Estimation over Mixture Weights

On the other hand, OBS can be seen as performing empirical Bayes on the mixture weights themselves. To see this, we interpret $\boldsymbol{\psi}$ as $\mathbf{w}$, and the posterior predictive density

$$p(y_i|\mathbf{y}_{1:i-1}, \mathbf{w}) = \sum_k w_k p_k(y_i|\mathbf{y}_{1:i-1}), \tag{22}$$

is the weighted mixture of the predictive densities of the individual models. The objective function that is being maximized in stacking in Eq. (6) is

$$\sum_{i=1}^{t} \log \left( \sum_k w_k p_k(y_i|\mathbf{y}_{1:i-1}) \right) = \sum_{i=1}^{t} \log p(y_i|\mathbf{y}_{1:i-1}, \mathbf{w}) = \log p_{\text{stack}}(\mathcal{D}_t|\mathbf{w}). \tag{23}$$

Hence, in stacking, we obtain the weights by maximizing the log-evidence of the 'stacking' model, $\log p_{\text{stack}}(\mathcal{D}_t|\mathbf{w})$, where the weights correspond to hyperparameters. Note the difference between this marginal likelihood and the one in BMA, where the marginal likelihood is itself a mixture and each component is independent of $\mathbf{w}$.

## 5 Experiments

Thus far, we have primarily focused on the theoretical properties of OBS. In this section, we provide empirical evidence that OBS can be beneficial. We consider four main scenarios: an illustrative toy example (Section 5.1), variational Bayesian neural networks (Section 5.2), SMC-based stochastic volatility models (Section 5.3), and dynamic Gaussian processes in non-stationary environments (Section 5.4). We provide details on our experimental setup, baselines, and code in Appendix B.

### 5.1 Subset Linear Regression

We revisit a classical problem from Breiman (1996), also used by Yao et al. (2018), generating i.i.d. data from a 15-dimensional Gaussian linear model with weak predictors, as in Yao et al. (2018, Section 4.2). We consider two scenarios with an ensemble of 15 regression models. In the **"Open" setting**, the ensemble consists of 15 univariate models ($y \approx \theta_k x_k$), none of which match the true process, where performance depends on combining models. In the **"Closed" setting**, model $k$ uses features $x_1, \ldots, x_k$, making model 15 the true model. In both scenarios, hyperparameters were pre-trained using empirical Bayes on 1000 points, followed by online deployment of OBS or O-BMA for 5000 points. Further experimental details are in Appendix C.

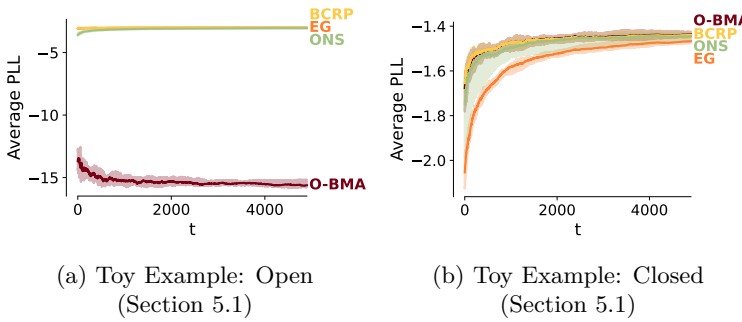

(a) Toy Example: Open
(Section 5.1)

(b) Toy Example: Closed
(Section 5.1)

Figure 1: The average predictive log-likelihood (higher is better) in the toy example. 'EG" is exponentiated gradients, "ONS" is the Online Newton Step, "BCRP" is the optimal constant rebalanced portfolio (offline baseline), and "O-BMA" is O-BMA. Lines denote the median and shaded area represent the 10th to 90th percentiles over 10 trials. The first 100 samples are suppressed for readability.

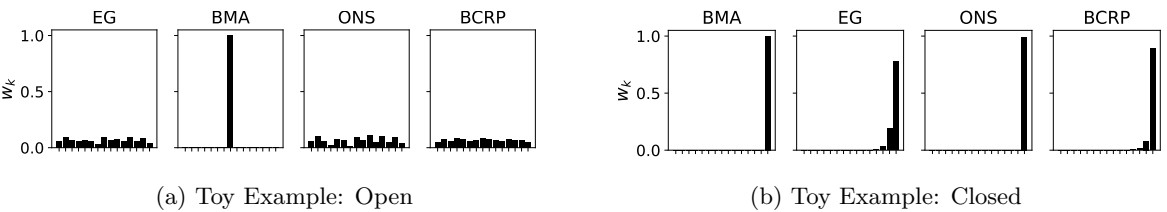

(a) Toy Example: Open

(b) Toy Example: Closed

Figure 2: The final weights in the "open" and "closed" subset linear regression experiment.

As shown in Figures 1a and 1b, OBS variants (EG, ONS) dramatically outperform O-BMA in the "open" setting. In the "closed" setting, OBS performs nearly as well as the theoretically optimal O-BMA, with ONS slightly outperforming EG.

As detailed in the theoretical section, one reason we may expect O-BMA to perform worse than OBS is the weight collapse of BMA. We can empirically validate this property in our toy experiment, with the expectation that O-BMA collapses to a single weight, and OBS retains a proper "mix" of models.

In Figure 2a, we show the final weights in the "open" subset linear regression experiment for O-BMA, OBS, and the BCRP (i.e., the optimal pooling of Geweke & Amisano (2011), or Bayesian stacking from the prequential principle (Yao et al., 2018)). In Figure 3a, we show the evolution of the BMA and OBS weights as new data arrive. Finally, in Figure 2b and Figure 3b, we show the analogous final weights and the evolution of weights for the "closed" subset linear regression experiment.

These results show promising evidence of our approach: OBS converges to a set of weights very similar to the best retrospective weights within 5000 samples in both the "open" and "closed" variations. OBS with ONS seems to exhibit more of the "collapsing" than OBS with EG, but these differences could be due to the hyperparameter choices, which were set to the values, $\eta = 10^{-2}$ for EG, and $\delta = 0.8$, $\eta = \beta = 10^{-2}$ for ONS. We further observe the collapse of O-BMA, and that OBS can also "collapse" if appropriate.

## 5.2   Online Variational Inference

We now move to a more practical application in Bayesian machine learning, applying OBS to online Bayesian neural networks. We use the recently proposed Bayesian online natural gradient (BONG), which optimizes the expected log-likelihood with online mirror descent (Jones et al., 2024). Many different variants of BONG and related approaches are tested, unifying them under a common framework; we make online predictions on MNIST (LeCun et al., 2010) using the best variant tested, "`bong-dlr10-ef_lin`". To create ensembles, we train five Bayesian feedforward neural networks with different initializations. We provide more details on the experimental setup in Appendix D.

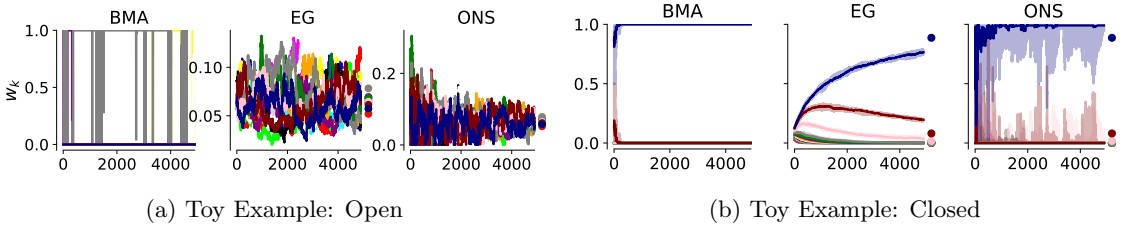

(a) Toy Example: Open                        (b) Toy Example: Closed

Figure 3: The evolution of the weight vector $\mathbf{w}_t$ as a function of $t$ in the "open" and "closed" subset linear regression experiment. Results are shown for a single trial due to the noisy nature of the plots. Dots on the right side of a plot denote the final weights of the BCRP.

We visualize the resulting predictive log-likelihoods in Figure 4a. Although less dramatic than in the toy example, we observe clear improvements in predictive performance using OBS instead of O-BMA and DMA. Furthermore, the specific OCO algorithm used does not seem to matter much, with Soft-Bayes, EG, and ONS all resulting in solutions similar to the BCRP. In Appendix D, we additionally visualize the evolution and final weights, which show that BMA collapses to a single model, whereas OBS converges to the optimal, more balanced weights.

### 5.3  Online Forecasting

We compare BMA and OBS variants for forecasting S&P data, which were used in an offline analysis by Geweke & Amisano (2011; 2012), ensembling diverse GARCH models. Model parameters are estimated online using sequential Monte Carlo (Doucet et al., 2001), where the typical "bank of filters" approach is equivalent to BMA. Eight models (2 per class, with priors sampled from uniform hyperpriors) were generated – see Appendix E for further details.

We visualize the resulting predictive log-likelihoods in Figure 4b. Again, we observe clear improvements in the predictive performance of OBS over O-BMA, with similar results obtained by Soft-Bayes, EG, and ONS. As this is real-world financial data, we expect some amount of non-stationarity: indeed, we find that DMA improves upon BMA, but remains inferior to the OBS methods, and that D-ONS performs the best. We show that BMA collapses once more in this example in Appendix E, which can be seen in the evolution and final weights.

Table 1: Median reward (predictive log-likelihood; higher is better) at the final time step in the GARCH experiment with 100 models.

| Method | Median Reward |
|---|---|
| EG | 3.46 |
| BMA | 3.44 |
| ONS | 3.46 |
| Soft-Bayes | 3.31 |

We additionally performed our online forecasting experiments with 25 models from each class, for a total of 100 models. Repeating across 3 random seeds, we obtain the results in Table 1. We used an EG learning rate 10 times larger than that used in previous experiments, guided by the logarithmic increase in the optimal theoretical learning rate as the number of models increases. The results show that ONS and EG still outperform BMA, while Soft-Bayes performs worse. We posit that the issues with Soft-Bayes are due to the relatively short time horizon (on the order of 1000 time steps) relative to the number of models – we are fundamentally trading off some fast adaptation for robustness when obtaining the gradient bound-free regret bounds of Soft-Bayes.

### 5.4  Online Regression in Non-Stationary Environments

A common issue in the online setting is non-stationarity, possibly due to covariate shift or concept drift. To illustrate the effectiveness of OBS in this setting, we apply OBS to the dynamic online ensemble of basis expansions (DOEBE) (Waxman & Djurić, 2024), which uses O-BMA to ensemble several online Gaussian process-based models. The DOEBE employs a linear basis approximation to GPs with random Fourier features (Lázaro-Gredilla et al., 2010; Rahimi & Recht, 2007) and models non-stationary processes by imposing

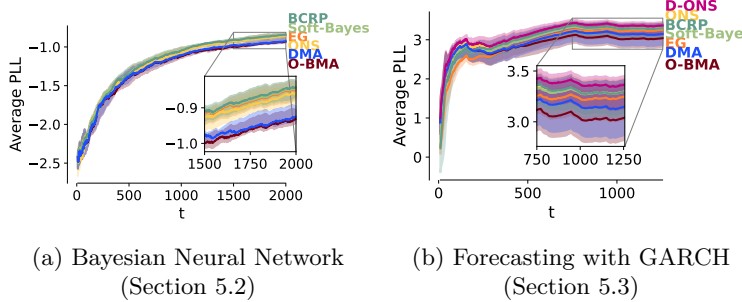

(a) Bayesian Neural Network
(Section 5.2)

(b) Forecasting with GARCH
(Section 5.3)

Figure 4: The average predictive log-likelihood (higher is better) in the MNIST and forecasting experiments, respectively. The method descriptions follow those in Figure 1, with the addition of Soft-Bayes. Lines denote the median and shaded area represents the 10th to 90th percentiles over 10 trials. The first 100 samples are suppressed for readability.

Table 2: The average predictive log-likelihood (median with 10th and 90th percentiles) in the non-stationary experiments; bolded values are best, and underlined values are second-best.

| Method Type | Method | Elevators | SARCOS | Kuka #1 | CaData | CPU Small |
|---|---|---|---|---|---|---|
| Online Baselines | O-BMA | $-0.59^{-0.58}_{-0.60}$ | $0.77^{0.78}_{0.74}$ | $0.61^{0.62}_{0.52}$ | $-0.84^{-0.83}_{-0.85}$ | $0.33^{0.34}_{0.32}$ |
| | DMA | $-0.59^{-0.58}_{-0.59}$ | $0.79^{0.80}_{0.77}$ | $0.69^{0.70}_{0.63}$ | $-0.81^{-0.80}_{-0.81}$ | $0.33^{0.34}_{0.33}$ |
| Offline | BCRP | $\mathbf{-0.57}^{-0.57}_{-0.58}$ | $0.80^{0.81}_{0.77}$ | $0.66^{0.68}_{0.62}$ | $-0.81^{-0.81}_{-0.81}$ | $\mathbf{0.35}^{0.37}_{0.35}$ |
| Online Bayesian Stacking | EG | $\underline{-0.58}^{-0.57}_{-0.58}$ | $0.78^{0.79}_{0.59}$ | $0.56^{0.67}_{0.45}$ | $-0.81^{-0.81}_{-0.81}$ | $0.35^{0.36}_{0.30}$ |
| | Soft-Bayes | $-0.58^{-0.58}_{-0.58}$ | $0.80^{0.81}_{0.77}$ | $\underline{0.70}^{0.72}_{0.67}$ | $-0.81^{-0.81}_{-0.81}$ | $0.34^{0.36}_{0.34}$ |
| | ONS | $-0.58^{-0.58}_{-0.59}$ | $\underline{0.80}^{0.82}_{0.78}$ | $0.70^{0.72}_{0.66}$ | $\underline{-0.80}^{-0.80}_{-0.81}$ | $0.34^{0.35}_{0.34}$ |
| | D-ONS | $-0.58^{-0.58}_{-0.58}$ | $\mathbf{0.81}^{0.82}_{0.79}$ | $\mathbf{0.73}^{0.74}_{0.69}$ | $\mathbf{-0.80}^{-0.80}_{-0.80}$ | $\underline{0.35}^{0.36}_{0.35}$ |

a random walk on these linear parameters, using variance $\sigma^2_{\mathrm{rw}}$. This parameter is set to a default value in Waxman & Djurić (2024), but it is found to be quite important for performance on several real-world datasets.

We apply OBS instead of O-BMA to ensemble RFF-GPs with $\sigma_{\mathrm{rw}} = 10^{-k}$, for $k \in \{0, 1, 2, 3, 4\}$. We use the same datasets as Waxman & Djurić (2024) (excluding purely synthetic ones); a summary of datasets and experimental details is in Appendix F. Of note are SARCOS and Kuka #1, both of which are robotics datasets with covariate shift. For SARCOS and Kuka #1 only, we use the smoothed version of EG (Helmbold et al., 1998) with hyperparameters $\eta = 10^{-3}$ and $\delta = 10^{-2}$; otherwise, EG exhibits severe instabilities.

We include numerical results in Table 2 and figures in Appendix F. We again find OBS to be beneficial. Though EG might be sensitive to outliers and occasionally performs poorly (particularly on SARCOS and Kuka #1), ONS and Soft-Bayes are rather robust and consistently outperform O-BMA. DMA again outperforms O-BMA, and D-ONS performs best on almost every dataset. Interesting future work includes analyzing adaptive regret in non-stationary settings.

## 5.5 Additional Experiments

We include several additional experiments in the appendices. Namely, in Appendix G, we perform a sensitivity analysis on the learning rate in OCO algorithms. We come to the general conclusion that the learning rate does not significantly affect the results of our experiments for a set of *a priori* sensible values.

Similarly, in Table 5, we include ablation on the forgetting factor hyperparameter in DMA. We find that more aggressive forgetting can help in highly non-stationary environments, but that OBS methods (and, in particular, D-ONS) remain competitive, even without further hyperparameter tuning.

# 6   Discussion & Conclusions

In this work, we critically study the idea of "Bayesian ensembles," especially in the online setting. We focused on two strategies for determining the weights in a linear ensemble: Bayesian model averaging (BMA) and Bayesian stacking (BS). In particular, we introduced and discussed OBS through the prequential principle. Through careful observation, we established a connection between the OBS problem and the OPS problem, allowing us to leverage the rich literature on online convex optimization. Below, we provide some practical guidance and limitations (Section 6.1), outlook (Section 6.2), and brief conclusions in Section 6.3.

## 6.1   Practical Guidance

**When to Prefer OBS.**   Our theory and experiments suggest that OBS is preferable in M-open scenarios, i.e., whenever the true data-generating model is not available, which we believe is typically the case in machine learning. When the problem is known to be M-closed, we reiterate previously-known theoretical results that O-BMA is optimal, though OBS, using many different online optimization algorithms, still provides competitive performance. Under anticipated and extreme non-stationarity, DMA with small forgetting factors is usually a viable approach, but otherwise, OBS using OCO algorithms designed for dynamic environments should be preferred.

**Choosing Your OCO Algorithm.**   The theoretical and practical properties of OBS depend on the OCO algorithm used. Based on our experiments and analysis of the relevant theory, we provide the following recommendations: **(1)** In scenarios with anticipated non-stationarities, D-ONS should be preferred. **(2)** In more general scenarios where the anticipated non-stationarity is modest and gradual, guidance is somewhat more subjective. Based on our empirical experiments, we recommend ONS if the computation is not prohibitive, EG if the computation is extremely prohibitive and extreme non-stationarity is not a concern, and Soft-Bayes otherwise; however, we encourage practitioners to experiment in their particular domains.

## 6.2   Outlook

We view OBS as providing a flexible framework, allowing for exciting research along two different axes.

First, the OBS connection provides an impactful application for new techniques developed in OCO. The connection also motivates a set of potentially interesting assumptions to incorporate into future regret bound analyses. For example, some recent work in OCO has focused on so-called "data-dependent bounds," which can provide more optimistic regret bounds depending on the statistics of the incoming data (Tsai et al., 2023; Putta & Agrawal, 2025); understanding the statistics of predictive log-likelihoods in certain online Bayesian learning settings could thus be a natural place to develop new bounds. Online learning also serves as a natural place to apply research on switching regret (Pasteris et al., 2024).

Second, OBS is quite a general framework for ensembling Bayesian models. Though we considered a wide range of models and applications in this paper (including variational Bayesian neural networks, GPs via basis expansion approximations, and stochastic volatility models via sequential Monte Carlo), we anticipate many more. For example, our sequential Monte Carlo experiment shows the benefits of OBS over the standard "bank of filters" approach to ensembling particle filters, suggesting novel applications in filtering theory.

It is also interesting to consider connections to other probabilistic machine learning settings, including non-Bayesian online deep learning methods (Valkanas et al., 2025), and applications of distributed or decentralized OCO (Yan et al., 2012; Mateos-Núnez & Cortés, 2014) to decentralized Bayesian inference methods that rely on O-BMA (Llorente et al., 2025).

## 6.3   Conclusions

Furthermore, we illustrate how O-BMA optimizes a different loss than OBS leading to a novel empirical Bayesian analysis, and show how O-BMA-based regret bounds can often be adapted to OBS. Finally, we empirically validate our theoretical claims with both synthetic and real datasets, showing that BMA collapses and that the proposed OBS algorithms deliver better performance.

The connection established in this work between OBS and the well-studied problem of portfolio selection bridges optimization and statistics and suggests many interesting future directions. For example, in online or continual learning, one is often concerned with properties under regime changes; a corresponding future direction is to evaluate OBS with algorithms aimed at minimizing *dynamic* or *adaptive regret* (Hazan, 2022, Chapter 10). Future statistical investigations include how to initialize ensembles well and whether the same rules of thumb apply as in BMA.

## Acknowledgments

This work was supported by the National Science Foundation under Award 2212506.

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

# A    More Details on Regret Analysis

One of the main benefits of connecting OBS with OPS is the numerous regret bounds available in the literature. Stating regret bounds for applications of OBS in general is not possible, as the available regret bounds depend on both the OCO algorithm and the class of models being ensembled. Below, we first state some general bounds reflecting the state of the art in OPS. We then provide an example of how these bounds may be applied, showing a regret analysis for OBS applied to an ensemble of approximate Gaussian processes.

## A.1    Regret Bounds from Portfolio Selection

Different choices of algorithm and different assumptions on the predictive densities $\mathbf{p}_t$ affect the obtainable regret analysis (c.f. Van Erven et al. (2020, Table 1)). However, what is generally possible in *streaming* algorithms (i.e., those whose runtime is linear in $T$) without making further assumptions on the loss is regret proportional to $\mathcal{O}(\sqrt{T})$.

By making further assumptions on the loss function — and in particular, assuming that a market variability parameter $\alpha$ exists — we can obtain even sharper bounds, often for simple algorithms. For example, let $G_p$ be a bound on the $p$-norm of the gradients (which follows from the existence of a bounded market variability parameter $\alpha$). Then EG guarantees regret proportional to $\sqrt{T}$ with an appropriate choice of learning rate (Helmbold et al., 1998, Theorem 4.1). ONS provides even better logarithmic regret (Agarwal et al., 2006, Theorem 1).

In the absence of a market variability parameter $\alpha$, both EG and ONS may be modified with a "smoothing" analysis, which artificially bounds the gradients, with the resulting EG regret bounds being proportional to $T^{3/4}$ (Helmbold et al., 1998, Theorem 4.2), and the resulting ONS regret bounds being proportional to $\sqrt{T \log T}$ (Agarwal et al., 2006, Theorem 1). Soft-Bayes can provide regret bounds independent of $G$ and without prior knowledge using time-varying learning rates, reducing dependence on $T$ to $\sqrt{T}$.

How to apply these bounds to achieve results for learning algorithms is an interesting question and depends on the learning algorithm and the type of bound to prove. Below, we provide a simple example that shows OBS may preserve the asymptotic properties of existing regret bounds, even when regret is measured with respect to a single expert rather than a mixture.

## A.2    Regret of Online BMA

We now prove Proposition 3.1.

**Proposition 3.1.** *Let the regret of the BMA mixture with respect to the best individual model be defined as*

$$R_T = \sum_t \log p_{k^*}(y_t \,|\, \mathbf{x}_t, \mathcal{D}_{t-1}) - \sum_t \log \left( \sum_k w_{t,k} p_k(y_t \,|\, \mathbf{x}_t, \mathcal{D}_{t-1}) \right), \tag{24}$$

*where $k_*$ is the best model. Then $R_T$ is related to an evidence gain in $M_{k^*}$,*

$$R_T = \log \frac{\Pr(\mathcal{M}_{k^*} \,|\, \mathcal{D}_T)}{\Pr(\mathcal{M}_{k^*})}. \tag{25}$$

*Proof.* This follows quickly from the definition of the O-BMA weights Equation (3). Denoting $\ell_t^{(\text{BMA})} = -\sum_t \log \left( \sum_k w_{t,k} p_k(y_t \,|\, \mathbf{x}_t, \mathcal{D}_{t-1}) \right)$ and $l_t^{(k^*)} = -\log p_{k^*}(y_t \,|\, \mathbf{x}_t, \mathcal{D}_{t-1})$, note that

$$\frac{w_t^{(k)}}{w_{t-1}^{(k)}} = \frac{\exp(-l_t^{(k_*)})}{\exp(-\ell_t)}.$$

Therefore, we arrive at a telescoping term,

$$\exp \left( \sum_{t=1}^T \ell_t^{(\text{BMA})} - l_t^{(k)} \right) = \frac{w_1^{(k_*)}}{w_0^{(k_*)}} \frac{w_1^{(k_*)}}{w_2^{(k_*)}} \cdots \frac{w_T^{(k_*)}}{w_{T-1}^{(k_*)}} = \frac{w_T^{(k_*)}}{w_0^{(k_*)}}.$$

Taking the logarithm and plugging in the closed form values Equation (2) proves the proposition.    □

### A.3 Example: Online Ensembles of Basis Expansions

A recent example of O-BMA being applied in machine learning is in the ensembling of (approximate) Gaussian processes (GPs). In particular, Lu et al. (2022) proposes ensembling approximate GPs with O-BMA, with Waxman & Djurić (2024) generalizing the algorithm and bounds to more general linear basis expansions. They both prove regret bounds relative to any expert with fixed parameters, a setting in which BMA is the appropriate optimizer.

It will first be useful to state the following lemma:

**Lemma A.1** (Kakade & Ng (2004), Theorem 2.2). *Let $\ell(\cdot; y_\tau)$ denote the negative log-likelihood, and assume it is $\mathcal{C}^2$ with second derivatives bounded by $c \in \mathbb{R}$. We are concerned with the negative predictive log-likelihood $\ell_t$ of a Bayesian linear model with basis expansion $\phi(\cdot)\colon \mathbb{R}^d \to \mathbb{R}^F$, using prior $p(\boldsymbol{\theta}) = \mathcal{N}(\boldsymbol{\theta}; \mathbf{0}, \sigma_{\boldsymbol{\theta}}^2 \mathbf{I}_F)$.*

*Let $\mathbf{x}_1, \ldots, \mathbf{x}_T$ be a sequence of inputs such that $\|\phi(\mathbf{x}_t)\|$ is bounded by 1 for all $t$. Then we have the following bound between the cumulative log-loss of the Bayesian estimator and the log-loss for any fixed value $\boldsymbol{\theta}_*$:*

$$\sum_{t=1}^{T} \ell_t - \ell\left(\phi(\mathbf{x}_t)^\top \boldsymbol{\theta}_*; y_t\right) \leq \frac{\|\boldsymbol{\theta}_*\|^2}{2\sigma_{\boldsymbol{\theta}}^2} + \frac{F}{2} \log\left(1 + \frac{Tc\sigma_{\boldsymbol{\theta}}^2}{F}\right).$$

In terms of the time horizon, the bound promised by Lemma A.1 is clearly $\mathcal{O}(\log T)$ in $T$.

We now formally state and prove the theorem.

**Theorem A.2.** *Let the negative log-likelihood $\ell(\cdot; y_\tau)$ be $\mathcal{C}^2$ with second derivatives bounded by $c \in \mathbb{R}$. We then consider an online ensemble of basis functions (Waxman & Djurić, 2024) with basis expansions $\phi^{(k)}\colon \mathbb{R}^{d_X} \to \mathbb{R}^{F_k}$ and priors $p(\theta^{(k)}) = \mathcal{N}(\boldsymbol{\theta}^{(k)}; \mathbf{0}, {\sigma_{\boldsymbol{\theta}}^{(k)}}^2 \mathbf{I}_{F_k})$ for $k \in \{1, \ldots, K\}$. Further, assume that $\|\phi^{(k)}\|$ is bounded by 1.*

*We will consider the log-loss of the ensemble at some pre-selected time $T$, denoted $\ell_T$, and its regret with respect to the performance of any single expert $k$ with a fixed parameter $\boldsymbol{\theta}_*^{(k)}$. Then **(a)** using O-BMA, the resulting regret is $\mathcal{O}(\log T)$ in $T$; **(b)** if we further assume that the log-loss is bounded, then the resulting regret remains $\mathcal{O}(\log T)$ using OBS with ONS as the optimizer.*

*Proof.* The result **(a)** is proved directly in Waxman & Djurić (2024, Theorem 1), which is adapted from Lu et al. (2022, Lemma 2). What remains to show is **(b)**. To emphasize the similarity in proving the O-BMA and OBS results, we will present them side by side.

Let $l_t^{(k)}$ denote the log-loss of the $k$th expert at time $t$. The proof then proceeds in two steps: first, bounding the loss of the ensemble estimate to any individual expert, and then applying Lemma A.1 and combining the bounds.

**Bounding Ensemble Losses to Experts** We proceed by first bounding the ensemble loss to the loss of any individual expert. Beginning with O-BMA, and proceeding identically to Proposition 3.1, Lu et al. (2022) make the observation that

$$\frac{w_{t-1}^{(k)}}{w_t^{(k)}} = \frac{\exp(-\ell_t)}{\exp(-l_t^{(k)})}.$$

Therefore, using initial weights $w^{(k)} = 1/K$,

$$\exp\left(-\sum_{t=1}^{T} \ell_t^{(\text{BMA})} + l_t^{(k)}\right) = \frac{w_0^{(k)}}{\cancel{w_1^{(k)}}} \frac{\cancel{w_1^{(k)}}}{\cancel{w_2^{(k)}}} \cdots \frac{w_{T-1}^{(k)}}{w_T^{(k)}} = \frac{1}{M w_T^{(k)}}.$$

Thus, we can bound the regret of O-BMA with any individual expert as

$$\sum_{t=1}^{T} \ell_t^{(\text{O-BMA})} - l_t^{(k)} \leq \log M.$$

For OBS, we apply the bound of Hazan et al. (2007, Theorem 2), yielding

$$\sum_{t=1}^{T} \ell_t^{(\text{OBS})} - l_t^{(k)} \le \sum_{t=1}^{T} \ell_t^{(\text{OBS})} - \left( \sum_m w_m^* l_t^{(k)} \right) \le \mathcal{O}(\log T),$$

where Big O notation absorbs constants related to $M$ and the maximum gradient norm.

**Bounding Expert Loss** Now, we bound the cumulative loss $\sum_{t=1}^{T} l_t^{(k)}$ to the loss using the fixed parameter $\boldsymbol{\theta}_*^{(k)}$. This is achieved by applying Lemma A.1 to the $k$th expert, which results in

$$\sum_{t=1}^{T} l_t^{(k)} - \ell \left( \phi^{(k)}(\mathbf{x}_t)^\top \boldsymbol{\theta}_*; y_t \right) \le \frac{\|\boldsymbol{\theta}_*\|^2}{2\sigma_{\boldsymbol{\theta}}^{(k)2}} + \frac{F_k}{2} \log \left( 1 + \frac{Tc\sigma_{\boldsymbol{\theta}}^{(k)2}}{F_k} \right) \in \mathcal{O}(\log T).$$

Combining the two bounds completes the proof. $\qquad\square$

# B  Experimental Setup

In this appendix, we discuss our experimental setup, including the OCO algorithms and hyperparameters used throughout.

## B.1  Experimental Setup

All experiments were conducted using Ubuntu 22.04 with an Intel i9-9900K CPU with 128 GB of RAM and two NVIDIA Titan RTX GPUs. Code for our methods and experiments is available online under an MIT License. [2] Implementations were in Jax/Objax (Bradbury et al., 2018; Objax Developers, 2020)[3] based on modifying the codes of Waxman & Djurić (2024)[4] and Jones et al. (2024)[5]. Optimization algorithm implementations were adapted from and tested against the Universal Portfolios library (Vinkler, 2024)[6].

## B.2  Baselines

**Best Constantly Rebalanced Portfolio** The BCRP is the so-called "static" solution to our stacking problem, equivalent to the offline versions of Bayesian stacking presented in Hall & Mitchell (2007); Geweke & Amisano (2011). The resulting weights are a solution to Equation (6), i.e.,

$$\mathbf{w}^* := \arg\max_{\mathbf{w} \in \mathbb{S}^K} \sum_{t=1}^{T} \log \sum_{k=1}^{K} w_k p_k(y_t \mid \mathbf{x}_t, \mathcal{D}_{1:t-1}). \tag{26}$$

In regression problems without concept drift, this forms a reasonable baseline against which to measure regret.

**Online Bayesian Model Averaging** Our main baseline is O-BMA, which we consider to be the standard online ensembling approach in Bayesian applications. The weights are updated as Equation (3), i.e.,

$$w_{t+1,k}^{\text{BMA}} = \frac{w_{t,k} p_k(y_{t+1} \mid \mathbf{x}_{t+1}, \mathcal{D}_t)}{\sum_k w_{t,k} p_k(y_{t+1} \mid \mathbf{x}_{t+1}, \mathcal{D}_t)}. \tag{27}$$

---

[2] https://github.com/DanWaxman/OnlineBayesianStacking, MIT License
[3] https://github.com/google/objax, Apache 2.0 License
[4] https://github.com/DanWaxman/DynamicOnlineBasisExpansions, MIT License
[5] https://github.com/petergchang/bong/tree/main, MIT License
[6] https://github.com/Marigold/universal-portfolios, MIT License

**Dynamic Model Averaging** In potentially non-stationary environments, we also compare to dynamic model averaging (DMA) (Raftery et al., 2010), which is essentially O-BMA with a forgetting factor $\gamma \in (0,1)$:

$$w_{t+1,k}^{\text{DMA}} = \frac{w_{t,k}^{\gamma} p_k(y_{t+1} \mid \mathbf{x}_{t+1}, \mathcal{D}_t)}{\sum_k w_{t,k}^{\gamma} p_k(y_{t+1} \mid \mathbf{x}_{t+1}, \mathcal{D}_t)}. \tag{28}$$

While DMA loses some of the nice statistical properties of O-BMA, it can be more robust in non-stationary scenarios, where the best expert may change over time. For the forgetting factor, we follow Raftery et al. (2010) and use a value of $\gamma = 0.99$.

### B.3 Portfolio Selection Algorithms

We include several OPS algorithms in our comparisons. We provide brief overviews of each and our default set of hyperparameters, below.

**Exponentiated Gradients** The EG algorithm (Helmbold et al., 1998) for portfolio selection follows the updates in Equation (10), i.e.,

$$\mathbf{w}_{t+1} = \frac{\mathbf{w}_t \odot \exp(\eta \mathbf{r}_t / \mathbf{w}_t^\top \mathbf{r}_t)}{\|\mathbf{w}_t \odot \exp(\eta \mathbf{r}_t / \mathbf{w}_t^\top \mathbf{r}_t)\|_1}, \tag{29}$$

where $\eta$ is a learning rate parameter. We choose a default value of $\eta = 10^{-2}$ in our experiments.

**Soft-Bayes** The Soft-Bayes algorithm (Orseau et al., 2017) provides several different potential updates, with regret guarantees independent of a market variability parameter $\alpha$. For our purposes, one useful formulation is the "online" variant of Orseau et al. (2017, Section 6), which updates weights using a time-varying learning rate $\eta_t$ as

$$w_{t+1,k} = w_{t,k} \left(1 - \eta_t + \eta_t \frac{r_t^k}{\sum_k w_{t,k} r_{t,k}}\right) \frac{\eta_{t+1}}{\eta_t} + \left(1 - \frac{\eta_{t+1}}{\eta_t}\right) w_{0,k}. \tag{30}$$

Square-root regret is then guaranteed (Orseau et al., 2017, Theorem 10) with the learning rate

$$\eta_t = \frac{\log K}{2Kt}, \tag{31}$$

which we use in our experiments.

**Online Newton Step** In our experiments, we use the form of ONS specialized for the OPS problem in Agarwal et al. (2006). This algorithm requires parameters $\eta, \beta, \delta$ and keeps track of quantities $\mathbf{A}_t$ and $\mathbf{b}_t$, defined as

$$\mathbf{A}_t = \sum_{\tau=1}^{t} -\nabla^2 \left[\log(\mathbf{w}_\tau \cdot \mathbf{r}_t)\right] + \mathbb{I}_K; \tag{32}$$

$$\mathbf{b}_t = \left(1 + \frac{1}{\beta}\right) \sum_{\tau=1}^{t} \nabla \left[\log(\mathbf{w}_\tau \cdot \mathbf{r}_t)\right]. \tag{33}$$

Weights are then obtained at each iteration by projecting onto the simplex,

$$\mathbf{w}_{t+1} = \Pi_{\mathbb{S}^K}^{\mathbf{A}_t} \left(\delta \mathbf{A}_t^{-1} \mathbf{b}_t\right), \tag{34}$$

where the projection operator $\Pi_{\mathbb{S}^K}^{\mathbf{A}_t}$ is defined as

$$\Pi_{\mathbb{S}^K}^{\mathbf{A}_t}(\mathbf{v}) = \arg\min_{\mathbf{w}} (\mathbf{w} - \mathbf{v})^\top \mathbf{A}(\mathbf{w} - \mathbf{v}). \tag{35}$$

Additionally, to help with very small market variability parameters, gradients may be artificially "smoothed" as

$$\tilde{\mathbf{w}}_t = (1 - \eta)\mathbf{w}_t + \frac{1}{K}\mathbf{1}. \tag{36}$$

We use default values of $\delta = 0.8$ and $\beta = \eta = 10^{-2}$.

**Discounted Online Newton Step**   The D-ONS algorithm (Yuan & Lamperski, 2020; Ding et al., 2021) is similar to the ONS algorithm (as in Hazan et al. (2007)), but it uses a forgetting factor for the second-order information. We use the formulation of Ding et al. (2021), which the authors claim is more numerically stable than the discounting factor used in Yuan & Lamperski (2020). The D-ONS algorithm depends on parameters $\eta > 0$ and $\gamma \in (0, 1)$ and keeps track of a quantity

$$P_{t+1} = (1 - \gamma)P_0 + \gamma P_t - \nabla_t^2 \left[\log(\mathbf{w}_\tau \cdot \mathbf{r}_t)\right]. \tag{37}$$

Updates are performed as

$$\mathbf{w}_{t+1} = \Pi_{\mathbb{S}^K}^{P_t} \left( \mathbf{w}_t - \frac{1}{\eta} P_t^{-1} \nabla \left[\log(\mathbf{w}_\tau \cdot \mathbf{r}_t)\right] \right). \tag{38}$$

We use default values of $\eta = 1.0$ and $\gamma = 0.99$.

## C   Details and Weights for the Subset of Linear Regressors Experiment

In this appendix, we provide more details on the experimental setup and results for the subset of the linear regression experiment in Section 5.1.

### C.1   Experimental Details

**Data Generation**   Following Breiman (1996) and Yao et al. (2018, Section 4.2), we generate 6000 data points i.i.d. according to:

$$\mathbf{x}_t \sim \mathcal{N}(5 \cdot \mathbf{1}_{15}, \mathbf{I}_{15}),$$
$$y_t \mid \mathbf{x}_t \sim \mathcal{N}(\mathbf{x}_t^\top \boldsymbol{\theta}, 1).$$

The ground truth parameters $\boldsymbol{\theta} \in \mathbb{R}^{15}$ are generated such that all 15 components are individually weak predictors, and the signal-to-noise ratio is 0.8; refer to Yao et al. (2018, Section 4.2) for the precise procedure used to generate $\boldsymbol{\theta}$.

**Model Descriptions**   We consider two ensembles, each comprising $K = 15$ Bayesian linear regression models:

- **"Open" setting:** Model $k$ ($k = 1, \ldots, 15$) is a univariate regression attempting to learn the function $y = \theta_k x_k$, using only the $k$-th component of $\mathbf{x}_t$.

- **"Closed" setting:** Model $k$ ($k = 1, \ldots, 15$) uses the first $k$ components of $\mathbf{x}_t$, i.e., attempting to learn $y = \sum_{j=1}^{k} \theta_j x_j$. Model $k = 15$ uses all features and corresponds to the true data-generating family.

Standard conjugate priors were used for the parameters $\theta_k$ (or vectors thereof) in each model.

**Hyperparameter Specification**   For both scenarios, the first 1000 data points were used to set the hyperparameters of the base linear regression models (e.g., prior variances) via empirical Bayes (type-II maximum likelihood on the marginal likelihood over these 1000 points). The subsequent 5000 points were processed online. This was accomplished by modifying the code of Waxman & Djurić (2024).

The OCO hyperparameters for OBS were set to reasonable default values and were not tuned further:

- Exponentiated Gradients (EG): Learning rate $\eta = 10^{-2}$.

- Online Newton Step (ONS): Parameters $\delta = 0.8$, $\eta = 10^{-2}$, $\beta = 10^{-2}$.

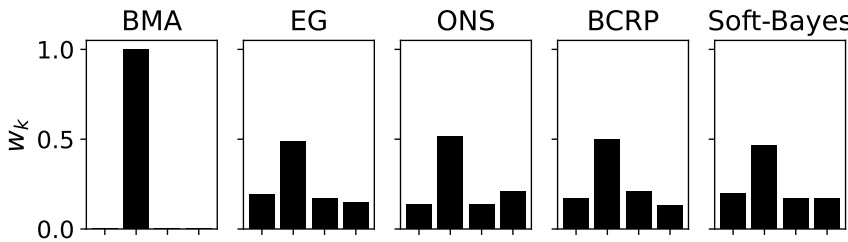

Figure 5: The final weights in the online variational inference experiment.

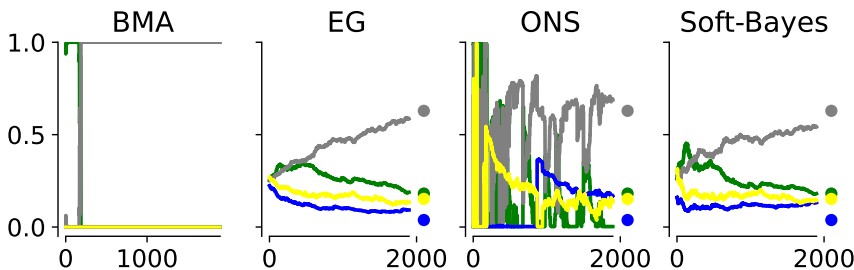

Figure 6: The evolution of the weight vector $\mathbf{w}_t$ as a function of $t$ in the online variational inference experiment. Results are shown for the last trial. Dots on the right side of a plot denote the final weights of the BCRP.

## D    Details and Weights for the Online Variational Inference Experiment

In this appendix, we provide more details on the experimental setup and results for the online variational inference experiment in Section 5.2.

### D.1    Experimental Details

As mentioned in the main text, we use the `bong-dlr10-ef_lin` model from Jones et al. (2024), which had the best performance among the variants tested in their experiments. The BONG performs variational inference using natural gradients, updating variational parameters $\boldsymbol{\psi}_t$ as

$$\boldsymbol{\psi}_{t+1} = \boldsymbol{\psi}_t + \mathbf{F}_{\boldsymbol{\psi}_t}^{-1} \nabla_{\boldsymbol{\psi}_t} \mathbb{E}_{\boldsymbol{\theta}_t \sim q_{\boldsymbol{\psi}_t}} [\log p(\mathbf{y}_t \mid \mathbf{x}_t, \boldsymbol{\theta}_t)],$$

where $\mathbf{F}_{\boldsymbol{\psi}_t}$ is the Fisher information matrix and $q_{\boldsymbol{\psi}_t}$ is the variational posterior. The `bong-dlr10-ef_lin` variant uses a variational family of Gaussians with low-rank covariance matrices (diagonal + a rank 10 matrix) and approximates the predictive likelihood through a linearization using the empirical Fisher information matrix. We use the authors' implementation of BONG.[7]

Our experimental setup generally mirrors that of Jones et al. (2024), with the following exceptions: we use a feedforward neural network with layers of width 64 and 32. The prior mean is sampled from the default `flax` initialization, and we form an ensemble over the prior variances $\sigma_0^2 \in \{10^{-2}, 10^{-1}, 10^0, 10^1\}$. Each "trial" corresponds to a different random shuffling of the training data, with the first 2000 data points used for inference.

### D.2    Weight Evolutions and Final Weights

Once again, we visualize the resulting weights in Figure 5 and Figure 6. We find a remarkable similarity between the final weights in all OCO algorithms tested and the BCRP solution, and that once again, BMA incorrectly collapses to a single model.

---

[7]https://github.com/petergchang/bong/, MIT License

# E  Details and Weights for the Forecasting Experiment

In this appendix, we provide more details on the experimental setup and results for the forecasting experiment in Section 5.3.

## E.1  Experimental Details

In the forecasting experiment, we use real data corresponding to the daily returns of the S&P 500 index from 2015 through 2020, consisting of 1257 unique observations. In econometrics, such data are often modeled using Generalized Autoregressive Conditional Heteroskedasticity (GARCH) approaches, which provide probabilistic predictions for time series. We refer the reader to Engle (2001) for a more detailed introduction to GARCH models.

**Specification of Models**  In Geweke & Amisano (2011; 2012), the application of GARCH models to stock index data is also used to test the ensembling of probabilistic models. In GARCH, the conditional variance of the process is modeled as an autoregressive process depending on the lagged values of the conditional variance and the squared "innovations." For instance, the GARCH$(1,1)$ model with Gaussian innovations is given by:

$$y_t = \sigma_t \epsilon_t, \ \ \epsilon_t \sim \mathcal{N}(0,1) \tag{39}$$

$$\sigma_{t+1}^2 = \alpha_0 + \alpha_1 \epsilon_t^2 + \beta \sigma_t^2, \tag{40}$$

where $\epsilon_t$ are the innovations and $\sigma_t$ is the conditional variance of the process. For brevity, we only discuss the GARCH$(1,1)$ with Gaussian innovations. The variants differ in the assumed process for the conditional variance.

We assign truncated Gaussian prior densities to $\boldsymbol{\theta}_t = [\alpha_0, \alpha_1, \beta, \sigma_t]$, and we are interested in sequential estimation of the posterior distribution $p(\boldsymbol{\theta}_t|y_{1:t})$. Sequential Monte Carlo (SMC) algorithms are very well suited for this task Doucet et al. (2001). An SMC algorithm recursively computes a particle approximation of the posterior,

$$p(\boldsymbol{\theta}_t|y_{1:t}) \approx \sum_i \rho_t^{(i)} \delta(\boldsymbol{\theta}_t^{(i)} - \boldsymbol{\theta}_t), \tag{41}$$

where $\boldsymbol{\theta}_t^{(i)}$ and $\rho_t^{(i)}$ denote, respectively, the particles and weights. After the arrival of $y_{t+1}$, the weights are recomputed, and the particles are propagated to form the particle approximation of the posterior at $t+1$. Using this particle approximation, we can obtain an approximation of the posterior predictive density,

$$p(y_{t+1}|y_{1:t}) \approx \sum_i \rho_t^{(i)} p(y_{t+1}|y_{1:t}, \boldsymbol{\theta}_{t+1}^{(i)}), \tag{42}$$

where $\boldsymbol{\theta}_{t+1}^{(i)}$ only differs from $\boldsymbol{\theta}_t^{(i)}$ in the component $\sigma_{t+1}^{(i)}$, which is obtained by substituting the previous set of particles in Eq. (40).

We create ensembles using several different variants of the GARCH model, namely GARCHt (Student-t innovations), GARCHNormal (Normal innovations), GJR-GARCHNormal (GJR-GARCH with Normal innovations), and EGARCHNormal (EGARCH with Normal innovations) Engle (2001).

**Inference and Hyperparameter Specification**  With each variant, we sample two sets of hyperparameters and perform online posterior inference using an SMC algorithm with 1000 particles and 5 Markov Chain Monte Carlo (MCMC) rejuvenation steps. At each iteration, the ensemble weights are used to evaluate,

$$\sum_{k=1}^8 w_{k,t} p_k(y_{t+1}|y_t) \approx \sum_{k=1}^8 w_{k,t} \sum_i \rho_t^{(i)} p_k(y_{t+1}|y_{1:t}, \boldsymbol{\theta}_{t+1}^{(i)}), , \tag{43}$$

where $w_{k,t}$ is the weight of model $k$ obtained at time $t$ using the approaches discussed in the paper. We ran 10 independent simulations of this experiment, considering 10 different random seeds.

### E.2 Weight Evolutions and Final Weights

We visualize the resulting weights in Figure 7 and Figure 8.

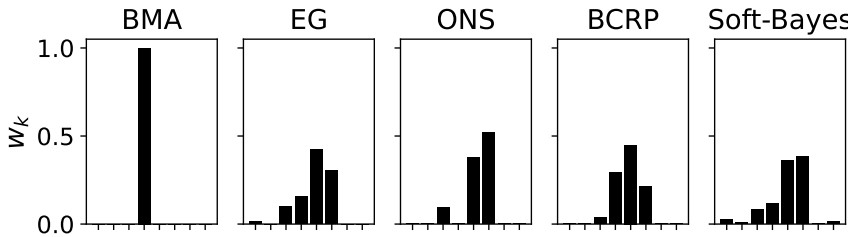

Figure 7: The final weights in the forecasting experiment.

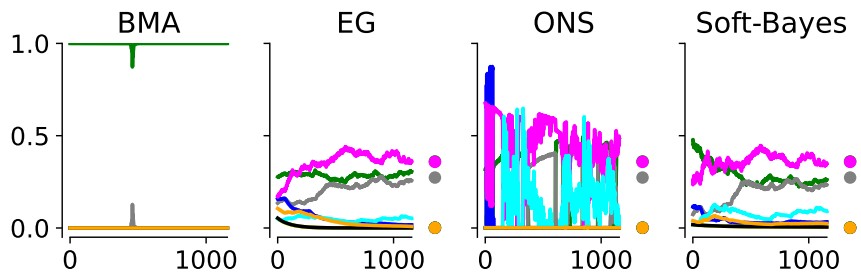

Figure 8: The evolution of the weight vector $\mathbf{w}_t$ as a function of $t$ in the forecasting experiment. Results are shown for the last trial. Dots on the right side of a plot denote the final weights of the BCRP.

## F    Details and Weights for the Non-Stationary Environment Experiment

In our experiment in non-stationary environments, we use the real or semi-real datasets in Waxman & Djurić (2024). This comprises the Elevators dataset (Torgo, 2024), which originates from tuning elevators on an aircraft; the SARCOS dataset (Rasmussen & Williams, 2005), which is simulated data corresponding to an inverse kinematics problem on a robotic arm; the Kuka #1 dataset (Meier et al., 2014), which is real data from a task similar to SARCOS; CaData (Pace & Barry, 1997) which has housing data, and CPU Small (Delve Developers, 1996), which includes various performance properties in a database of CPUs.

As mentioned in the main text, the models considered all belong to the *dynamic online ensembles of basis expansions* (DOEBE) family, which perform online GP regression using Kalman filtering and the random Fourier feature approximation Lázaro-Gredilla et al. (2010). Waxman & Djurić (2024) find that adding a random walk to the model parameters is important in capturing "dynamic" (i.e., nonstationary) behavior when applying approximate GPs to several real-world datasets, which can be related to "back-to-prior forgetting" (Van Vaerenbergh et al., 2012).

We create ensembles using an RFF GP with 100 features, trained on the first 1000 data points using the marginal likelihood. We then use different values of the random walk scale ($\sigma_{\mathrm{rw}} \in \{10^{-4}, 10^{-3}, 10^{-2}, 10^{-1}, 10^{0}\}$) and ensemble them using O-BMA and various OBS algorithms.

In the interest of space, we only report the weight evolutions for SARCOS, Kuka #1, and Elevators in Figure 10. The other dataset results are qualitatively similar to Elevators.

### F.1 Cumulative Reward Plots

We show the cumulative reward plots (i.e., the average PLL as a function of $t$) in Figure 9.

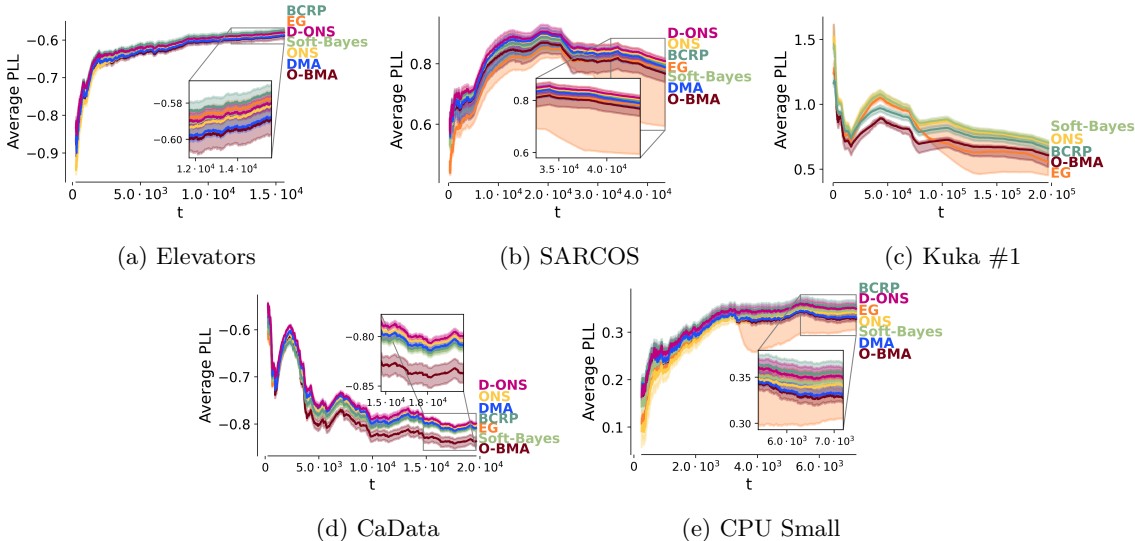

Figure 9: The average predictive log-likelihood (higher is better) in the non-stationary experiments. Lines denote the median and shaded area represents the 10th to 90th percentiles over 10 trials. The first 250 samples are suppressed for readability.

## G   Sensitivity to the Learning Rate

The learning rate marks an important hyperparameter in online convex optimization, with regret bounds often being quite sensitive to the learning rate. In this section, we investigate the practical effects of the learning rate parameter in EG and ONS.

For the experiment using the BONG (Section 5.2 & Appendix D), we performed additional experiments using differing hyperparameters in OCO algorithms. Specifically, we vary $\eta$ in EG and $\beta$ in ONS for values $10^{-k}$ and $k \in \{0, 1, 2, 3, 4\}$. Results can be found in Table 3. We find that the results are not particularly sensitive to the learning rate, with all results except for EG with a high learning rate ($10^0$) and ONS with a small learning rate ($10^{-4}$) outperforming O-BMA.

Results are similar in the forecasting example (Section 5.3 & Appendix E), where the results in Table 4 suggest largely comparable performance across learning rate values. The performance of EG degrades at very large or very small learning rates, but ONS seems fairly stable in this setting.

## H   Dynamic Model Averaging With Differing Forgetting Factors

In our experiments in Section 5.4, we used a default "forgetting factor" for DMA, as recommended in Raftery et al. (2010). This is similar to our other experiments, in which we leave OBS hyperparameters at reasonable defaults and do not otherwise tune them. Nevertheless, to further understand the impacts of this hyperparameter, we repeated our experiments with more aggressive and more conservative values of the forgetting factor. We report results in Table 5.

Overall, we find that DMA with extremely low forgetting factors performs well in highly nonstationary environments (Kuka #1, CaData), but it is still comparable to D-ONS, which has more robust guarantees regarding predictive performance. Moreover, this is without tuning the hyperparameters of D-ONS. In other datasets tested, OBS methods are still superior to DMA (CPU Small, SARCOS, Elevators). We additionally remark that DMA has much similarity with a sliding window approach and loses many of the statistical properties of O-BMA; on the other hand, adaptive/dynamic OCO algorithms D-ONS still come with robust theoretical guarantees.

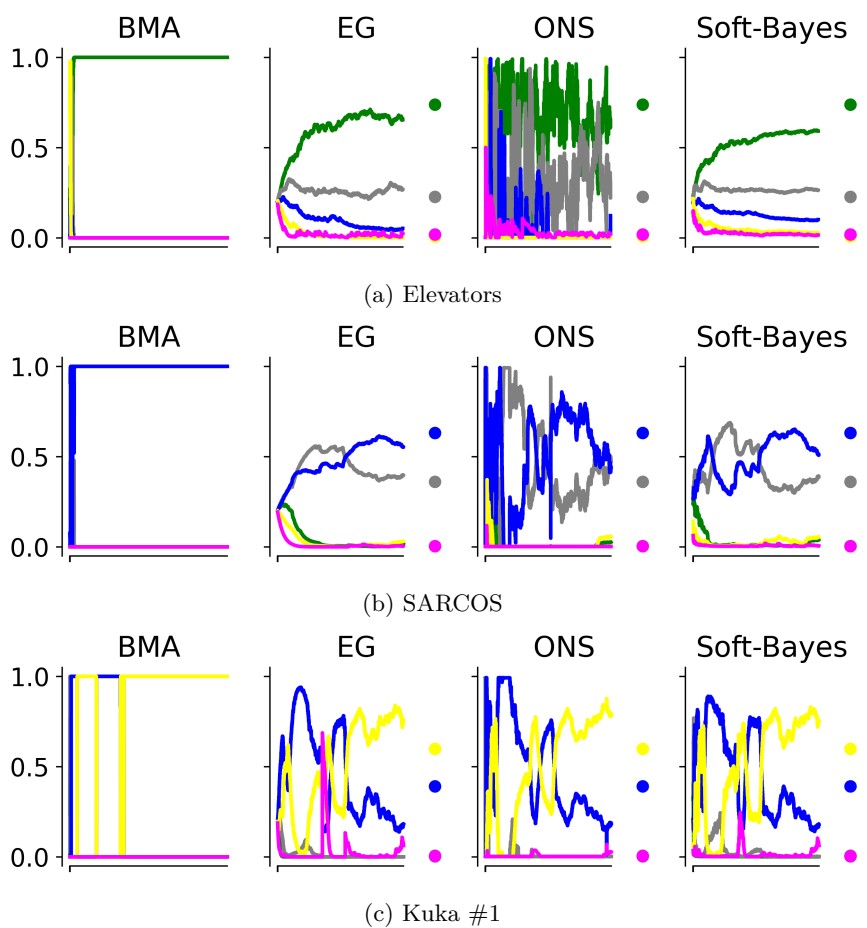

Figure 10: The evolution of the weight vector $\mathbf{w}_t$ as a function of $t$ in the non-stationary environment experiment in the Elevators, SARCOS, and Kuka #1 datasets. Results are shown for the first trial. Dots on the right side of a plot denote the final weights of the BCRP.

Table 3: Comparison of the average predictive log likelihood value (higher is better) for the BONG experiment (Section 5.2) while varying the learning rates of EG and ONS. Results are reported as the median $\pm$ 1 standard deviation across five trials.

| Method | Learning Rate | Average PLL $\pm$ Std. Dev. |
|---|---|---|
| EG ($\eta$) | $10^0$ | $-1.012 \pm 0.04$ |
| | $10^{-1}$ | $-0.850 \pm 0.02$ |
| | $10^{-2}$ | $-0.850 \pm 0.02$ |
| | $10^{-3}$ | $-0.862 \pm 0.02$ |
| | $10^{-4}$ | $-0.865 \pm 0.02$ |
| ONS ($\beta$) | $10^0$ | $-0.878 \pm 0.03$ |
| | $10^{-1}$ | $-0.859 \pm 0.03$ |
| | $10^{-2}$ | $-0.860 \pm 0.02$ |
| | $10^{-3}$ | $-0.900 \pm 0.03$ |
| | $10^{-4}$ | $-0.948 \pm 0.03$ |
| Soft-Bayes | $-$ | $-0.847 \pm 0.02$ |
| O-BMA | $-$ | $-0.932 \pm 0.02$ |
| BCRP | $-$ | $-0.843 \pm 0.02$ |

Table 4: Comparison of the average predictive log likelihood value (higher is better) for the GARCH experiment (Section 5.3) while varying the learning rates of EG and ONS. Results are reported as the median $\pm$ 1 standard deviation across ten trials.

| Method | Learning Rate | Average PLL $\pm$ Std. Dev. |
|---|---|---|
| EG | $10^0$ | $2.776 \pm 0.23$ |
| | $10^{-1}$ | $3.188 \pm 0.13$ |
| | $10^{-2}$ | $3.249 \pm 0.13$ |
| | $10^{-3}$ | $2.978 \pm 0.14$ |
| | $10^{-4}$ | $2.858 \pm 0.17$ |
| ONS | $10^0$ | $3.256 \pm 0.12$ |
| | $10^{-1}$ | $3.330 \pm 0.13$ |
| | $10^{-2}$ | $3.300 \pm 0.13$ |
| | $10^{-3}$ | $3.229 \pm 0.10$ |
| | $10^{-4}$ | $3.159 \pm 0.12$ |
| Soft-Bayes | – | $3.296 \pm 0.13$ |
| O-BMA | – | $3.040 \pm 0.24$ |
| BCRP | – | $3.296 \pm 0.12$ |

Table 5: Comparison of methods on non-stationary datasets (c.f. Table 2), including DMA with additional forgetting factors.

| Method | Elevators | SARCOS | Kuka #1 | CaData | CPU Small |
|---|---|---|---|---|---|
| O-BMA | -0.59 | 0.77 | 0.61 | -0.84 | 0.33 |
| DMA | -0.59 | 0.79 | 0.69 | -0.81 | 0.33 |
| DMA-0.9 | -0.60 | 0.80 | **0.75** | -0.80 | 0.33 |
| DMA-0.95 | -0.59 | 0.80 | 0.73 | -0.80 | 0.33 |
| EG | -0.58 | 0.78 | 0.56 | -0.81 | 0.35 |
| Soft-Bayes | -0.58 | 0.80 | 0.70 | -0.81 | 0.34 |
| BCRP | **-0.57** | 0.80 | 0.66 | -0.81 | **0.35** |
| ONS | -0.58 | 0.80 | 0.70 | -0.80 | 0.34 |
| D-ONS | -0.58 | **0.81** | 0.73 | -0.80 | 0.35 |

