# OpenReview forum: "Bayesian Ensembling: Insights from Online Optimization and Empirical Bayes"
_TMLR — Accepted by TMLR_

### Review · Reviewer_xoWH · 2025-10-18

**Summary Of Contributions:**

This paper studies combining Bayesian predictive models in an online setting by
casting the stacking objective as a portfolio-selection problem. Building on
online convex optimization, it proposes Online Bayesian Stacking (OBS) with
practical updates (e.g., Exponentiated Gradient, Soft-Bayes, Online Newton
Step), provides an empirical Bayes interpretation explaining why BMA tends to
collapse to a single model, and leverages regret bounds from the portfolio
literature. Numerical experiments are carried out to show that OBS
outperforms online BMA and dynamic model averaging.

**Audience:**

Yes

**Audience Explanation:**

Obtaining the optimal weights for Bayesian model averaging is challenging, and
effective methods are needed. This paper proposes to treat the problem as
portfolio selection to leverage existing algorithms.

**Claims And Evidence:**

Yes

**Claims Explanation:**

The portfolio selection problem is a very general framework and it seems
straightforward to apply it to model averaging and selection problems.

**Requested Changes:**

Putting model averaging weights optimization in the framework of
portfolio selection seems straightforward. Please explain the novelty or
challenge of this approach.

A key reason why the optimal weights are challenging to obtain is that they sum
to one and are non-negative. This constraint makes the optimization problem hard
to solve. But I don't see how the portfolio selection framework helps with
this. Please explain.

Why $w$ is $K-1$ dimensional simplex? Shouldn't it be $K$ dimensional simplex?

There is a large body of literature on determining the weights of model
averaging, both in the Bayesian and frequentist settings. They are mostly
ignored by the authors. Please discuss the related literature and how the
proposed method compares with them.

Proofread the paper for consistency in notations and terminologies. For example,
it refers Proposition 3.1 as "the following lemma" and Corollary 3.2 as "the
following theorem".

What do you mean by "model parameters are integrated out" in pgae 7 above (16)?

It is difficult to distinguish different methods in the figures on gray scale.

---

> ### Author Response · Authors · 2025-12-14
>
> We thank the reviewer for their review and questions, which we address below. All revisions have been incorporated, which we will post by the end of the day tomorrow (Sunday, 12/14).
>
> ---
>
> ## On Novelty and Utility
>
> > Putting model averaging weights optimization in the framework of portfolio selection seems straightforward. Please explain the novelty or challenge of this approach.
>
> While we understand that the mathematical connection is not complex, we still believe there to be significant value in making such connections. In particular, we do not believe this connection has been made explicitly in the literature before, and in particular, no empirical study has been performed to illustrate improved performance. We find this important, as until now, the standard approach in online Bayesian ensembles has been O-BMA; in some ways, we think it more important to demonstrate that a simple but effective alternative exists over a complex one.
>
> Moreover, we provide significant analysis extending the simple connection between OPS and OBS. Methodologically, we present OBS as a flexible framework whose algorithms posses distinct tradeoffs. We further provide a novel empirical Bayes perspective that clarifies the benefits of stacking from a statistical perspective.
>
> ---
>
> ## Additional Questions
>
>  > A key reason why the optimal weights are challenging to obtain is that they sum to one and are non-negative. This constraint makes the optimization problem hard to solve. But I don't see how the portfolio selection framework helps with this. Please explain.
>
> This is exactly the benefit of using the OCO framework: the convex constraint of the simplex is well-studied within this parallel literature, and not well-utilized for online learning in Bayesian ensembles. Rather than developing new algorithms from scratch and re-solving a classical problem from optimization, we are able to directly exploit decades of work from the optimization community for this difficult problem. For example, the EG algorithm arises from a regularization step to online gradient descent meant to make the simplex constraint more tenable, and the ONS algorithm is derived from online 2nd-order optimization, with the analysis for OPS specialized to deal with the simplex constraint.
>
> > Why $w$ is $K-1$ dimensional simplex? Shouldn't it be $K$ dimensional simplex?
>
> Because of the constraint that $\sum w_k = 1$, the final weight $w_K$ is actually determined exactly by the remaining $K-1$ weights, i.e., $w_K = 1 - \sum_{k=1}^{K-1} w_k$. There are thus $K - 1$ degrees of freedom.
>
> > There is a large body of literature on determining the weights of model averaging, both in the Bayesian and frequentist settings. They are mostly ignored by the authors. Please discuss the related literature and how the proposed method compares with them.
>
> We attempt to discuss at length potential alternatives, including but not limited to Section 2 of the paper, resulting in over 60 references. While we are happy to discuss specific additional references the reviewer may have in mind, we are not aware of particular references that we felt added to the discussion of online Bayesian ensembles. We wish to stress our desire to both accommodate any additional references relevant to our work, but also that our work is not meant to be a review paper.
>
> > What do you mean by "model parameters are integrated out" in pgae 7 above (16)?
>
> We mean that they are marginalized, in the standard Bayesian way.

---

### Review · Reviewer_8MRG · 2025-10-21

**Summary Of Contributions:**

The authors study the ensembling, via linear combination, of bayesian supervised learning models in an online setting, i.e., where models are updated after each new labelled data point is ingested. The authors present Online Bayesian Stacking (OBS) , where the combination of models in the ensemble is chosen to optimize a log-predictive score. OBS is contrasted with O-BMA (online bayesian model averaging), in which individual models in the ensemble are weighted in proportion to the evidence that the model generated the data. A correspondence between OBS and online portfolio selection (OPS) from finance theory is demonstrated. O-BMA and OBS are also shown to be versions of empirical Bayes w.r.t. indicator variables and mixture weightes, respectively. Various algorithmic subtypes of OBS (exponentiated gradients, soft bayes, etc) are studies. Superior results for OBS as compared to O-BMA are obtained on a toy problem, an online version of MNIST, and S+P forecasting problem and non-stationary ensembling of gaussian processes. A rule of thumb is proposed where O-BMA may be preferable in the "closed" setting where the true data-generating model is among the models in the ensemble, but OBS performs better in the (much more common) "open" scenario where the true model is not in the ensemble.

**Audience:**

Yes

**Audience Explanation:**

I don't think the audience for this paper is very broad, but it's broad enough to merit TMLR's consideration.

**Broader Impact Concerns:**

No concerns here.

**Claims And Evidence:**

Yes

**Claims Explanation:**

I think this paper may be able to get to the point where I could say 'yes' with some revisions, but I don't think it's there yet.

The authors mostly use single, default settings for hyperparameters in their experiments. I would like to see some comparisons where each method has undergone a hyperparameter optimization (or at least, would like to hear an explanation for why that's not feasible or appropriate).

I would also like more detail about how to interpret and use the distinction between the 'open' and 'closed' setting when picking OBS vs. O-BMA. Common sense would dictate that there is actually a continuum here and I would presume that O-BMA could still be preferable if something very, very close to the ground truth model is available. E.g., if a model in the ensemble could achieve r-squared of 0.999, I would presume that O-BMA could still beat OBS. If that's not the case, I would like to hear why. It's hard to believe that anything even very slightly less that a perfectly closed setting means that OBS wins. I would also like to hear more about how to assess whether we are in the open or closed setting.   How does one distinguish between being in the open setting vs. just dealing with a problem with lots of inherent noise in the outcome variable? If the perfect model could only get r-squared 0.5 because of noise, how are you going to distinguish, in a practical way, between open and closed ?

**Update after rebuttal**

In light of the open vs. closed discussions and the pointer to the hyperparameter experiments, I have changed my 'claims' answer to 'Yes'.

**Requested Changes:**

As indicated above, I would like to see experiments with hyperparameter optimization and I would like much more elaboration on open vs. closed (whether or not it's a continuum where O-BMA still wins for nearly closed and also how to detect open vs. closed).

Some suggested edits/typos:

2nd paragraph page 3 “in this case, can compute” -> "in this case, we can compute"

In section 2.2, I would note that the second set is a 'validation set' or 'holdout set'  so that the terminology is more familiar to ML researchers.

I'm not sure what 'contributed discussion' means in section 2.3. Maybe re-write that.

On page 5, in the 'simplest OCO' paragraph, I would not use the variable x, since it has a different meaning elsewhere in the paper. I would only refer to the w instead of explaining that x corresponds to w.

---

> ### Author Response · Authors · 2025-12-14
>
> We thank the reviewer for their thoughtful review of our work. Below, we address the reviewer's concerns point-by-point. All revisions have been incorporated, which we will post by the end of the day tomorrow (Sunday, 12/14).
>
> ---
>
> ## Regarding Hyperparameter Ablation
>
> We would like to point the reviewer to Appendix G: Sensitivity to the Learning Rate, and Appendix H: Dynamic Model Averaging With Differing Forgetting Factors of the original submission, which we believe were unfortunately not highlighted well. We have revised the manuscript to mention the existence of these appendices more prominently, and believe they should quell the reviewer's concerns.
>
> In Appendix G, we vary the learning rate of OBS methods in the online variational inference and online forecasting experiments, with the main finding that, within a reasonable set of hyperparameter values, the resulting methods do not differ substantially in performance, and the main takeaways from the main manuscript hold.
>
> In Appendix H, we select multiple different hyperparameter values in the competing method of dynamic model averaging (DMA). We find that with rather aggressive forgetting factors, DMA performs well in highly nonstationary environments, but even in this case, D-ONS still performs well (without additional hyperparameter tuning).
>
> We additionally note that hyperparameter tuning is generally difficult in online settings, as we typically lack a proper validation set, and may anticipate undetermined distribution shifts.
>
> If the reviewer's concerns persist, we are happy to provide additional experiments, though we are unsure the exact form they would take and are open to suggestions.
>
> ---
>
> ## On the distinction between the "open" and "closed" settings
>
> Thank you for raising this interesting point; while "open" and "closed" are tautologically binary under our framework --- where M-closed means that the data generating process lies within our potential models, and M-open means that it does not --- we agree that there are varying degrees to which model classes may be M-open.
>
> Importantly, our claim is not that OBS strictly dominates O-BMA whenever the setting departs even slightly from closure. Indeed, when the ensemble contains a model that is extremely close to the ground truth, for example, one achieving near-perfect predictive performance, O-BMA can still be competitive and may outperform OBS. In such cases, the posterior concentration underlying BMA effectively identifies the dominant model, and the benefit of adaptive weighting diminishes. The advantage of OBS becomes pronounced as model misspecification increases and combinations of models begin to outperform single models. In this regime, O-BMA tends to over-concentrate on the least misspecified model, whereas OBS can leverage complementary predictive strengths across models and adapt more effectively to evolving data distributions.
>
> One main point which we wish to reinforce is that even in the M-closed setting, where O-BMA is optimal, OBS still obtains vanishing regret, and empirically converges to the correct ground truth weighting (i.e., one-hot weighting on the "true" model). This is reflected in the "subset linear regression" experiment of Section 5.1, where ONS provides results competitive with O-BMA, and converges to the "correct" final weights (c.f., Fig. 2(b)).
>
> ## On assessing if we are in the open or closed setting.
>
> We thank the reviewer for raising this important point. We agree that, in practice, the distinction between an open and a closed setting cannot be determined *a priori*, and that limited predictive performance may arise either from irreducible noise or from model misspecification. In our framework, the open/closed distinction is not intended as a binary classification that must be resolved before applying the method. Rather, it is a conceptual framework for interpreting model behavior. A closed setting corresponds to the existence of a model in the ensemble whose residual error is dominated by irreducible noise, whereas an open setting reflects systematic discrepancies that persist even after accounting for noise. Practically, these regimes can only be assessed indirectly.
> As noted in the Dynamic Model Averaging framework of Raftery et al. (2010), whether a problem behaves as effectively M-closed or M-open cannot be decided in advance, but can be inferred indirectly from the evolution of posterior model weights: stable concentration signals effective closure, while persistent weight adaptation reflects model misspecification.
>
> One interesting method to accomplish this is the use of O-BMA to hierarchically ensemble the O-BMA and OBS solutions. Based on the corresponding regret analysis, in the M-closed setting the result maintains constant regret. We've added a small section explaining this in the revised manuscript.
>
> ## Minor Changes
>
> We thank the reviewer for the helpful editorial suggestions and will implement them in the revised manuscript.

---

> > ### Comment · Reviewer_8MRG · 2026-01-02
> > **Thanks**
> >
> > Thank you for the clarifications regarding hyperparameter ablation and open vs. closed. Very helpful.

---

### Review · Reviewer_eozu · 2025-11-30

**Summary Of Contributions:**

The authors examine the problem of Bayesian ensembles in an online learning setting, where the goal is to select an optimal combination of known Bayesian prediction models to approximate the true data generator. Known methods for Bayesian ensembles include Bayesian model averaging (BMA) and its well-studied online extension, as well as Bayesian stacking (BS) which have showed improved performance when the true data generator is not derived from one of the Bayesian options.

This submission's main contribution is a novel studying of BS's extension into the online learning model (OBS), as well as the delicate observation that OBS corresponds to the well-studied online convex optimization problem, online portfolio selection (OPS). This observation allows the authors to connect established regret bounds and analytical methods from online convex optimization to the problem of Bayesian ensembles.

From this lens a novel theoretical analysis is also provided, explaining how BMA fails when faced with assumptions that BS can overcome. Empirical data is provided, illustrating the improved performance obtained by OBS over other ensembling methods under these assumptions.

**Audience:**

Yes

**Audience Explanation:**

Establishing connections from machine learning problems to classical theoretical problems such as online convex optimization is a very intriguing direction for research and can easily inspire future explorations along this line of thought, as outlined by the author's concluding sections. From a personal perspective I think the field of machine learning would benefit from more theoretically-backed results such as the ones presented in this paper.

**Broader Impact Concerns:**

Impact statements are not necessary for this submission.

**Claims And Evidence:**

Yes

**Claims Explanation:**

The submission makes mostly theoretical claims that is supported by proof and intuitions. I am especially fond of the extensive explanations given in the paper for the significance and novelty of their results - the introductory sections paint a clear picture of the prior works and background knowledge necessary for understanding the main contribution of this paper, that (in my opinion) is the connection between this machine learning problem of Bayesian ensembles and convex optimization.

**Requested Changes:**

- Page 3, paragraph 2: "In this case, **we** can compute..." where the "we" is missing.

---

> ### Author Response · Authors · 2025-12-13
>
> We sincerely thank the reviewer for their positive assessment of our work. In the revised version, we have fixed the mentioned typo.

---

### Author Response · Authors · 2025-12-15

We would like to once again thank all the reviewers for their constructive and helpful feedback. We have uploaded a revised version of the manuscript to help address reviewer concerns. We highlight the main changes below:

1. **[Hybrid Solutions for M-open or M-closed detection; 8MRG]** We have added a new section (Section 3.5) which discusses a hybrid solution, which maintains the expressivity of OBS while obtaining regret bounds that are a constant factor worse than O-BMA in the M-closed scenario.

2. **[Pointers to Additional Experiments; 8MRG]** We have added additional pointers to experiments in the appendices surrounding hyperparameter in Section 5.5.

3. **[Minor Edits; All Reviewers]** We have fixed the typos and minor editorial errors caught by the reviewers.

---

### Decision · Action_Editor_5nw6 · 2026-01-12

**Recommendation:** Accept as is

**Audience:**

Yes

**Audience Explanation:**

Yes -- ensembling is a widespread technique and novel methods in this area certainly would be of interest to some members of TMLR's audience.

**Claims And Evidence:**

Yes

**Claims Explanation:**

The reviewers universally agree that the claims made in the manuscript -- in particular, after revision following the discussion period -- are supported by accurate, convincing, and clear evidence. The reviewers mention that the paper is easy to read, and that the main contributions are well motivated and supported both in discussion and by empirical study.

During the initial review period, one reviewer did express some minor concerns regarding the strength of the claims in the paper, but after some discussion and revision, these concerns were overcome.